# Kinetics and Fluid-Specific Behavior of Metal Ions After Hip Replacement

**DOI:** 10.3390/bioengineering13010044

**Published:** 2025-12-30

**Authors:** Charles Thompson, Samikshya Neupane, Sheila Galbreath, Tarun Goswami

**Affiliations:** 1Department of Neuroscience, Cell Biology, and Physiology, Wright State University, Dayton, OH 45435, USA; thompson.853@wright.edu; 2Department of Biomedical, Industrial, and Human Factors Engineering, Wright State University, Dayton, OH 45435, USA; neupane.36@wright.edu (S.N.); galbreath.9@wright.edu (S.G.)

**Keywords:** arthroplasty, replacement, hip, hip prosthesis, serum, blood, urine, metals, heavy, cobalt, chromium, titanium, random forest, kinetics, transport, biological

## Abstract

**Background**: Total hip arthroplasty (THA) is a well-tolerated and effective procedure that can improve a patient’s mobility and quality of life. A main concern, however, is the release of metal ions into the body due to wear and corrosion. Commonly reported ions are Co and Cr, while others, such as Ti, Mo, and Ni, are less frequently studied. The objective of this study was to characterize compartmentalization and time-dependent ion behaviors across serum, whole blood, and urine after hip prosthetic implantation. The goal of using Random Forest (RF) was to determine whether machine learning modeling could support temporal trends across data. **Methods**: Data was gathered from the literature of clinical studies, and we conducted a pooled analysis of the temporal kinetics from cohorts of patients who received hip prosthetics. Mean ion concentrations were normalized to µg/L across each fluid and weighted by cohort sample size. RF was used as a study-level test of predictive accuracy across ions. **Results**: For serum and whole blood, Co and Cr displayed one-phase association models, while Ti showed an exponential rise and decay. Ions typically rose quickly within the first 24 months postoperatively. Serum Co and whole blood had similar patterns, tapering off just under 2 µg/L, but serum Cr (~2.02 µg/L) was generally higher than that of whole blood (~0.99 µg/L). Mean urinary Co levels were greater than those of Cr, suggesting a larger, freely filterable fraction for Co. RF was implemented to determine predictive accuracy for each ion, showing a stronger fit for Co (R^2^ = 0.86, RMSE = 0.57) compared to Cr (R^2^ = 0.52, RMSE = 0.50). **Conclusions**: Sub-threshold exposure was prevalent across cohorts. Serum and whole blood Co and Cr displayed distinct kinetic profiles and, if validated, could support fluid-specific monitoring strategies. We present a methodology for interpreting ion kinetics and show potential for machine learning applications in postoperative monitoring.

## 1. Introduction

Annually, more than 1 million total hip arthroplasties (THAs) are performed globally, with the United States projected to experience a 284% increase by 2040 [1,2]. As the number of procedures continues to increase, continued research in metal ion kinetics and compartmentalization is necessary to provide surgeons with objective data to integrate into their practice and meet the increasing need within the population [3].

Metal alloys are the material of choice due to their desirable material properties, such as electrochemical stability and biocompatibility [1,4]. The choice of material and implant type is determined based on factors such as location, surgeon preference, and patient-specific factors (e.g., age, bone quality, and activity), which can influence long-term performance. The alloys are selected based on the device components and their material properties, such as corrosion resistance, wear tolerance, and mechanical strength, making it ideal for load-bearing parts [2,5].

The main metal components of THA are composed of stainless steel (SS), titanium (Ti), cobalt–chromium (CC), CoCrMo, tantalum, or TiAlV [2,5]. Despite the SS corrosion tendency, the manufacturing of the device is straightforward, and the material has a slow oxidation rate [5]. The femoral stem and head are typically manufactured with cobalt–chromium (Co-Cr and Co-Cr-Mo) because of its greater Young’s modulus (220–230 Gpa), two times as much compared to Ti-Al-V alloy, and wear performance, when compared to Ti alloys (110–120 Gpa) [5,6]. The typical locations of Ti-6AL-4V are the stem and acetabular components; it has material properties that are both biologically compatible with bone and strong, but its wear resistance is not ideal [5]. A complete list of metal composition by percentage for orthopedic alloys is discussed in the Methods.

Heavy-metal ions are present in trace quantities in healthy human life and are needed for different cellular proteins throughout the body. For example, Cr is involved in the metabolism of glucose and aids in carbohydrate and lipid catabolism, while Co^3+^ is necessary in the catalytic site of vitamin B12, which is important for red blood cell production [7]. However, the role of Co and Cr in genotoxicity and cytotoxicity is generally related to their oxidation state [8,9,10]. Hexavalent chromium (Cr^6+^) is considered a group 1 carcinogen by the International Agency for Research on Cancer (IARC), while Co^2+^ salts and other Co compounds are in group 2 [11]. Since 2019, organizations such as the European Chemicals Agency (ECHA) and the International Agency for Research on Cancer (IARC) have categorized different forms of cobalt as either “probably carcinogenic to humans,” “possibly carcinogenic to humans,” or “may cause cancer in humans” [4]. It has been hypothesized that the ions released from the metal may be the mechanism of cytotoxicity and genotoxicity, and ion migration can result in additional adverse effects [4,12].

The release of metal ions at both the bearing surfaces and modular junctions may be a result of micromotion, the fretting impact of the passive oxide layers, thereby accelerating material degradation [3]. Metal ion generation has been linked to adverse local tissue reactions (ALTRs), adverse reactions to metal debris (ARMD), and other implant-failure-related mechanisms [13,14,15,16,17,18]. Systemic metal ion distribution in biological fluids also raises concerns about adverse neurological, cardiovascular, and endocrine effects [19,20]. Extensive mechanistic and pathophysiological details are provided in the Appendix A [6,21,22,23,24,25,26,27,28,29,30,31,32,33,34,35,36,37,38,39,40,41,42,43,44,45,46,47,48,49,50,51,52,53,54,55,56,57,58,59,60,61,62,63,64,65,66,67,68,69,70,71,72,73,74].

Considering that clinicians typically monitor serum, whole blood, and urine, the complete kinetic profile of metal ions is poorly understood. Research gaps exist due to inconsistent research methodologies and variations in follow-up periods across studies, which further complicate interpretation and leave gaps in how clinicians and researchers can effectively utilize these measures to anticipate risk. In addition to Co, Cr, and Ti, ions such as Ni and Mo are occasionally reported but remain poorly characterized in vivo. Many available studies are small and methodologically diverse, which shows the value of pooled synthesis in observing broad-scale temporal trends. The novelty of this research lies in applying machine learning tools to longitudinal data. This approach provides insight into the kinetic behavior of metal ions and their potential relevance for time-dependent implant surveillance and clinical monitoring.

This study synthesized data from the published literature to evaluate temporal patterns of Co, Cr, Ti, and Mo ions in serum, whole blood, and urine following hip prosthetic implantation. The objective was to characterize the time-dependent behavior of ions and compare their biological fluid distribution. In addition, RF was applied to determine whether machine learning models can support observed temporal trends when interpreting ion concentrations from the data. Defining the broad-scale kinetic trajectories of these ions may provide insight into reference patterns for clinicians to judge if lab tests reflect regular postoperative changes or signs of early pathology. Unusual persistence or extreme divergence from such patterns may indicate excessive wear, tribocorrosion, or other mechanisms driving ion release.

## 2. Methods

### 2.1. Disclosure of Scope

The current study provides a pooled synthesis of published data rather than raw patient-level data. Because reported values varied in statistical treatment and lacked consistent reporting of variance, all pooled analyses and models should be interpreted as exploratory. The primary aim of this work is to assess the generalizable patterns in large-scale, temporal changes in ion concentration, which could generate hypotheses for future prospective studies.

### 2.2. Surgical-Grade Alloy Chemical Composition

The metal ion data included in this analysis are from orthopedic implants manufactured using alloys with compositions standardized to ASTM and ISO specifications. To aid interpretation of which metallic implant materials are likely sources of ions, Table 1 describes the nominal chemical compositions of implant-grade alloys.

These references have been provided to support the interpretation of the pooled ion concentrations reported in clinical studies. For instance, the primary source of cobalt and chromium ions in these datasets is CoCr alloys. While other factors can alter ion release kinetics, such as implant design, surface finish, and in vivo environment, alloy composition provides a framework for interpreting trends.

### 2.3. Study Selection

Peer-reviewed studies reporting metal ion concentrations in human subjects who received a metallic hip prosthetic were considered for inclusion. Studies reporting ion concentration in at least one bodily fluid (serum, whole blood, or urine), indicating the postoperative timepoint of sample collection, provide implant characteristics, and studying adult human subjects were included. Studies that pooled multiple implant types or sizes were included if the cohort had a consistent bearing surface or modularity profile with at least one metal–metal interface. Studies of other joint prostheses (i.e., knee, ankle, or spine) were excluded unless the time-concentration profile for hip prostheses was separately reported. Studies were excluded if the subjects were described as having renal insufficiency or high occupational exposure to metals or if the cohort underwent revision for implant failure. Inclusion and exclusion criteria are described in Figure 1. Except for the case of urine, cross-sectional studies that reported mean follow-up rather than specific timepoints were avoided.

Because the data was extracted from cohorts, the reason for hip implantation varied. Most studies described degenerative joint disease as the primary reason for implantation rather than trauma-related incidents. Demographic reports across included studies were inconsistent. From the available literature, the average age ranged from around 45 to 71 years, with a BMI range of 23 to 31. Across numerous reports, patients had not received other metallic implants, and if this was reported, bilateral and multi-implant cohorts were excluded to minimize confounding. The goal was to isolate cohorts who received a single hip prosthetic.

Figure 1 depicts the identification, screening and eligibility, and inclusion of articles used in the current article. Articles were located through open-access journals and PubMed searches by using search terms such as “longitudinal, temporal, hip replacement, metal ions, cobalt, chromium, titanium, molybdenum, nickel, serum, whole blood, urine, etc.” This resulted in thirty-six studies published in a window between the years 1998 and 2024. Individual studies differed in which metals and fluids were reported, so the corresponding number of articles varied by ion and fluid type. For instance, most studies reported cobalt and chromium in serum or whole blood, compared to titanium, molybdenum, and nickel, which were less frequently reported. Reported ions, fluids, and postoperative follow-ups are summarized in Appendix B, Table A1.

### 2.4. Data Extraction and Standardization

Ion concentration values were manually extracted from published data tables if available. Values presented only graphically were manually extracted using PlotDigitizer v3.3.9 PRO (PORBITAL, USA) after calibrating the axes in the original figure. Figures within this manuscript were redrawn and reanalyzed by the authors, and no figures were directly imported.

A “cohort” was defined as a distinct patient group with separately reported outcome data, even if multiple cohorts were described within a single publication. Variability within included cohorts can be an inherent limitation of pooled analysis. Possible sources of such heterogeneity include different implant designs/manufacturers, bearing sizes, patient activity levels and age, and follow-up intervals. To reduce this variability, units were standardized to micrograms per liter (µg/L) and weighted by cohort sample size at each reported timepoint. The goal of these methods was to minimize the impact of small or large cohorts to obtain stronger, broad-scale representation. Since each published study contained a unique cohort from differing author groups and institutions (Table A1), duplicates were highly unlikely. However, it should be noted that overlap between publications could not be completely ruled out. Therefore, all models should be viewed as exploratory and can be used as a guide for future research.

Reported concentrations in other units (i.e., nmol/L, ng/mL) were converted to µg/L using either SI relationships or molecular-weight-based calculations, if necessary. Reported study sample sizes were used to weight timepoint-specific values where available. If the sample size at follow-up timepoints was not specifically reported, the total cohort size reported at the beginning of the study was used as an approximation. Extracted entries are outlined in Table A1, a summary of studies following hip implantation, and annotated with

Reference identification;Ion species;Fluid type (serum, whole blood, urine);Fluid sample Collection timepoint (month);Analytical method;Prosthetic Summary.

### 2.5. Statistical Modeling Approaches

#### 2.5.1. Nonlinear Regression Comparing Fluid Compartmentalization

##### Data Processing

Ion concentration values were analyzed separately by fluid type to retain the physiological context and avoid cross-fluid comparisons. Serum, whole blood, and urine values were pooled at reported timepoints and weighted by study sample size according to Equation (1):(1)$${\bar{x}}_{\mathrm{weighted}}\;\frac{\sum_{i=1}^{n}x_{i}\times \;N_{i}}{\sum_{i}^{n}Ni}$$
where

*x_i_* = reported ion concentrations from published study *i* (µg/L);*N_i_* = number of participants from cohort *i* at corresponding timepoint;*n* = number of cohorts contributing to that timepoint.

Urine data were less frequently reported in longitudinal form and were instead grouped into postoperative time phases:Preoperative (≤0 months);Early (1–23 months);Middle (24–47 months);Late (≥48 months).

##### Model Development

Whole blood and serum values for cobalt and chromium were modeled to capture the longitudinal time-concentration trajectory of each using a one-phase exponential association (Equation (2)) in GraphPad Prism version 10.6.1 (799) (GraphPad Software, LLC, Boston, MA, USA) in the form(2)$$Y\left(t\right)=Y_{0}+Span\;(1-e^{-Kt})$$
where $Y_{0}$ is the modeled concentration at time zero, *Span* = *Plateau* − $Y_{0}$, and *K* is the rate constant. From these variables, the time constant was derived as $\tau =\frac{1}{K}$ and half-time) as $t_{\frac{1}{2}}=\frac{ln(2)}{K}$. Time is defined as t in Equation (2). 

Serum titanium trajectories were modeled with an exponential rise-and-decay association (Equation (3)) from GraphPad Prism in the form(3)$$Y\left(t\right)=Y_{0}+A(e^{-k_{r}t}-e^{-k_{d}t})$$
where $Y_{0}$ is the baseline concentration at time zero; *A* is a scaling parameter; and *k_r_* and *k_d_* represent the rate constants for rising and decaying phases, respectively.

##### Model Visualization

For visualization, shaded regions around the central trajectory represent the SD of pooled values at each timepoint, reflecting variability across contributing cohorts. These intervals show that while modeled curves capture the general postoperative rise toward a plateau, many follow-up points demonstrated wider dispersion, indicating observed heterogeneity rather than uniform stabilization. By modeling the early postoperative rise, this method captures the dominant early kinetic trend while taking into account how long-term concentrations vary considerably across patients, which are not fully described by a single curve.

##### Nonlinear Model Evaluation

Nonlinear regression was performed in GraphPad Prism, and 95% confidence intervals were calculated by profile likelihood. A constraint of *K* > 0 was applied to ensure early monotonic increases consistent with the expected early rise in postoperative kinetics.

Model adequacy was assessed by inspecting residual distributions, QQ plots, and actual versus predicted values. Goodness-of-fit statistics are reported but should be interpreted with caution, as nonlinear regression applied to pooled study-level data does not account for underlying patient-level differences.

#### 2.5.2. Machine Learning (Random Forest)

##### Data Preprocessing

Predictive models were limited to serum because this fluid was most commonly reported, with larger cohort sizes, number of participants, and timepoints greater than 60 months, which allowed for greater robustness in both model training and testing. The dataset was obtained from serum measurements and imported from Microsoft Excel using the pandas 1.2.4 library in Python 3.9.2 and numpy 1.20.3. The raw dataset contained three primary variables: timepoint (months), number of participants, and serum ion (Co, Cr, Ti, Mo, Ni) concentrations (µg/L). To standardize column names, variables were renamed time_months, participants, and conc_ug_L. Only relevant columns were retained, and rows containing missing values were excluded to ensure data integrity.

##### Feature Selection and Target Variable

The predictive target (dependent variable) was the serum ion concentration (conc_ug_L). The independent variables (features) were as follows:Time (months): Representing the temporal progression of measurements.Participants: Representing the number of individuals contributing to the average concentration at each timepoint.

This allowed the model to capture both temporal changes and the variability in population size.

##### RF Model Development and Evaluation

Random Forest was selected due to its robustness in handling nonlinear relationships and its resistance to overfitting compared with single decision trees. An RF regression model was implemented using scikit-learn library 0.24.1. The dataset was split into training (80%) and testing (20%) subsets using stratified random sampling with a random seed of 42 to ensure reproducibility. The model was trained using 200 decision trees with scikit-learn’s default settings, striking a balance between computational efficiency and predictive performance while providing stable performance with a single split. A formal hyperparameter search was not performed, as adding hyperparameter tuning using a grid search with 5-fold cross-validation yielded lower test-set performance.

No. of training and testing datasets for each ion:Co = train = 64, test = 16, Total = 80;Cr = train = 72, test = 18, Total = 90;Ti = train = 52, test = 14, Total = 66;Mo = train = 28, test = 8, Total = 36;Ni = train = 14, test = 4, Total = 18. 

After training, predictions were generated for both the training and test sets. The test dataset was evaluated to determine the model’s performance. The evaluated metrics are the mean absolute error (MAE), mean squared error (MSE), root mean squared error (RMSE), and the coefficient of determination (R^2^). The Matplotlib library 3.9.4 was utilized to conduct a graphical evaluation through scatter plots of actual versus predicted temporal trends of concentrations, parity plots for observed versus predicted concentrations against a 45° reference line, and residual plots to determine prediction errors across concentration ranges, detecting bias or heteroscedasticity.

##### Visualization and Interpretation

To complement the statistical evaluation, several visualizations were employed:Actual vs. Predicted Scatter Plot: Displaying temporal trends of measured and model-predicted concentrations.Parity Plot: Comparing observed vs. predicted concentrations against a 45° reference line to assess accuracy.Residual Plot: Analyzing prediction errors across concentration ranges to detect bias or heteroscedasticity.

These plots were generated using the Matplotlib library to enhance interpretability and provide publication-quality figures.

## 3. Results

Section 3.1, Section 3.2, Section 3.3, Section 3.4, and Section 3.5 summarize the reported ion concentrations at specified follow-up intervals. These descriptive figures are intended to depict overall patterns seen in the literature (Table A1); this serves as a foundation for the machine learning modeling presented in Section 3.6.

### 3.1. Cobalt

#### 3.1.1. Serum and Whole Blood Cobalt Trends

Serum and whole blood cobalt concentrations quickly increased early and then appeared to level off at a more sustained value (Figure 2). The serum modeled curve increased from 0.18 µg/L to a plateau of 1.96 µg/L with a half-time of 3.9 months, and the whole blood curve increased from 0.45 µg/L to a nearly identical plateau of 1.96 µg/L with a half-time of 5.7 months. Therefore, the serum values tended to reach their plateau slightly quicker, while the whole blood values demonstrated a slightly longer increase. The model for serum showed wider SD bands during follow-up, and the whole blood trajectories were typically more consistent over time. The relative fit of the model was stronger for whole blood (R^2^ = 0.36, Sy.x = 0.79) than serum (R^2^ = 0.20, Sy.x = 1.22), consistent with these differences in SD. Table 2 shows the number of reported participants and contributing cohorts at each follow up interval.

#### 3.1.2. Urinary Cobalt

Urinary cobalt concentrations increased throughout the postoperative phases, with the highest levels observed beyond 48 months (Figure 3. Preoperative concentrations were low (mean 0.82 µg/L, SD 0.44, n = 103 participants from 3 cohorts). Concentrations rose during the early postoperative period (mean 9.46 µg/L, SD 9.21, n = 155 from 5 cohorts) and continued to increase in the middle phase (mean 15.86 µg/L, SD 11.44, n = 139 from 5 cohorts). Late-phase concentrations were elevated (mean 48.04 µg/L, SD 40.51, n = 103 from 3 cohorts), though variability was high, which reflects the variety of cohorts. The pooled data suggest sustained urinary cobalt elevation four years postoperatively, with an increasing spread in the later follow-up periods. The number of participants and cohorts at each time phase are provided in Table 3. 

### 3.2. Chromium

#### 3.2.1. Serum and Whole Blood Chromium Trends

Chromium concentrations in serum and whole blood followed distinct postoperative trajectories. In serum, the curve rose from 0.28 µg/L to a plateau of 2.02 µg/L with a half-time of 8.0 months. This reflects a slower approach to equilibrium compared with cobalt. Whole blood Cr started at 0.30 µg/L and plateaued at a lower value of 0.99 µg/L with a half-time of 2.4 months. Serum Cr levels increased about two-fold and reached higher sustained concentrations. Standard deviation was greater in serum across most follow-up periods, whereas whole blood showed more consistency around the central line. The number of participants and cohorts for both serum and whole blood at each follow up interval is provided in Table 4.

**Table 4 bioengineering-13-00044-t004:** Data coverage for pooled serum and whole blood Cr concentrations shown in Figure 4.

	Serum		Whole Blood	
Time (Months)	Reported Participants	Contributing Cohorts	Reported Participants	Contributing Cohorts
0	365	9	450	12
3	163	3	88	3
6	220	6	194	6
9	-	-	55	2
12	647	15	289	10
24	671	15	344	10
36	308	7	113	2
48	212	5	-	-
60	424	10	282	8
72	131	3	-	-
84	225	5	-	-
96	95	3	-	-
108	147	4	-	-
120	83	3	124	2

#### 3.2.2. Urinary Chromium

As seen in Figure 5, the preop phase, shows lower urinary Cr with mean values of 0.17 µg/L (SD 0.18, n = 103) preoperatively. During the early phase, the concentration increased to 1.31 µg/L (SD 0.73, n = 155) in the early postoperative period (1–23 months). Levels began to increase slightly in the middle interval (24–47 months; mean 1.87 µg/L, SD 1.05, n = 139), followed by a bigger rise in the late period (>48 months), where the pooled mean reached 12.79 µg/L (SD 10.56, n = 103). The late time phase also had a noticeable spread, which is seen in the wide SD error bars. This shows a general increase in urinary Cr excretion after implantation, with wider variation as time moves on. Table 5 provides the number of reported participants and contributing cohorts for urinary Cr at each time phase. 

### 3.3. Titanium

#### Serum Titanium Trends

Serum Ti increased to a peak that fell between 12 and 24 months, followed by a more gradual decline (Figure 6). The exponential rise-then-decay model ($\left[Y\left(t\right)=Y_{0}+A(e^{-k_{r}t}-e^{-k_{d}t})\right]$) had a baseline of 0.67 µg/L and rise-and-decay rate constants of kᵣ = 0.049 and k_d_ = 0.028. Goodness-of-fit statistics (R2 = 0.22, Sy.x = 0.78, sum of squares—932) are provided to be transparent because of the use of pooled cohorts. The limited reported participants and number of cohorts beyond 60 months may also make late-term interpretation more difficult (Table 6).

### 3.4. Nonlinear Model Diagnostics

Table 7 provides a summary of the one-phase association model outputs for serum and whole blood Co and Cr.

#### 3.4.1. Serum Cobalt

Serum Co residuals were around zero and did not show systematic trends across time (Figure 7). The QQ plot shows mild curvature away from the identity line, pointing toward deviation from normality, with many of the points falling in a narrow range. For the most part, the actual vs. predicted plot showed no signs of extreme scatter, with many values falling near the identity line. As expected, the plots showed variance but did fit the overall central tendency of the dataset, which supports the one-phase association model for serum Co.

#### 3.4.2. Whole Blood Cobalt

Whole blood Co residuals were centered around zero (Figure 8), and there were no strong systematic trends across time, which indicates that the model fit the trajectory. There was approximate alignment with the identity line for the QQ plot, while the residuals fell in a narrow range with deviation at the tails. There was limited scatter for the actual vs. predicted plot with clustered values near the identity line and expected variance. These plots support the adequacy of the fit model for whole blood Co.

#### 3.4.3. Serum Chromium

The residuals were distributed around zero without clear bias over time for serum chromium. Although there was an outlier at an intermediate follow-up of 48 months (Figure 9), the QQ plot showed systematic curvature away from its identity line and points toward deviations from normality. For the most part, the actual vs. predicted plot showed tight clustering of observed values near its identity line, especially at higher concentrations. These plots help support the one-phase association model in capturing the central trajectory of serum Cr concentrations while acknowledging variation with pooled clinical data.

#### 3.4.4. Whole Blood Chromium

The residuals for whole blood Cr were mostly centered near zero without extreme bias across time, indicating that the model tracked the temporal pattern without systematic over- or underestimation. The QQ plot showed reasonable alignment with the identity line, with only modest deviations at the tails, suggesting approximate normality of residuals. The actual vs. predicted plot demonstrated close clustering of observed values near the identity line, with one point falling outside the expected range, consistent with limited scatter overall. Taken together, Figure 10 supports the adequacy of the one-phase association to describe whole blood chromium concentrations while acknowledging residual variability within the pooled dataset.

#### 3.4.5. Serum Titanium

Summary of exponential rise and decay output is summarized in Table 8 Serum Ti residuals showed tight centering around zero and no strong evidence of systematic bias over time (Figure 11). However, the QQ plot showed points within a narrow range and showed a distribution close to vertical for the theoretical distribution. This was expected due to the cohort-level dataset. There was a close cluster of the actual vs. predicted plot along its identity line with very little scatter. Thus, these plots indicate that the fitted exponential rise-and-decay model was able to capture the overall central trajectory of the dataset.

### 3.5. Molybdenum

Due to the limited number of studies reporting longitudinal Mo concentrations, combined with heterogeneity in reporting formats, implant configurations, and follow-up intervals, no pooled values were calculated. Instead, individual study data were extracted and plotted to illustrate reported concentrations over time (Figure 12).

Cross-sectional concentrations and average follow-up time are summarized in Table 9. Mean serum molybdenum concentrations ranged from 0.83 to 0.97 µg/L over follow-up intervals of 24–108 months. These findings were reported for both MoM THA and MoM BHR devices, with values staying in a consistent range across studies and average follow-up times.

### 3.6. Nickel

Longitudinal data for Ni concentrations were limited. Dahlstrand et al. reported serum Ni concentrations of over 24 months in both MoM (n = 28) and MoP (n = 26) hip devices (Figure 13). In each group, their relative concentrations increased over time, with the MoM cohort demonstrating a nearly twofold rise from baseline to 24 months, with MoP falling slightly below. On the other hand, Figure 14 shows data from Savarino et al. across longer follow up intervals. 

Across these studies, serum Ni concentrations remained below 2.5 µg/L. In one study not presented, Newton et al. measured Ni in whole blood (n = 199) and plasma (n = 205) at an average of 72 months follow-up and recorded mean concentrations of 3.0 µg/L and 2.4 µg/L, respectively. Although not plotted due to limited longitudinal data, these findings were consistent with normal reference ranges (<40 nmol/L, approximately 2.34 µg/L).

### 3.7. Random Forest Machine Learning

#### 3.7.1. Cobalt

The Random Forest plot compares measured cobalt (blue) concentrations with Random Forest (red) predictions across timepoints (Figure 15). The predicted points closely follow the actual values, with good overlap across both low and high concentrations. Occasional underestimation is visible at peaks (~5 µg/L), but overall, the model captures temporal fluctuations accurately (R^2^ = 0.861).

The Co parity plot demonstrates strong alignment between predictions and actual values. Most points cluster tightly around the diagonal, with limited scatter, indicating that the RF produced balanced, unbiased predictions.

Cobalt residuals are centered tightly around zero with no clear pattern relative to predicted values. This indicates that the model errors are small, random, and unbiased, confirming statistical robustness. Such behavior matches the strong parity alignment and high R^2^.

#### 3.7.2. Chromium

For Cr, predicted concentrations track the temporal variation of observed data but with larger deviations than Co. While the general pattern is reproduced, the Random Forest model underestimates certain peaks (>5 µg/L) and overestimates some mid-range values (Figure 16). This variability reflects moderate predictive performance, supported by an R^2^ = 0.522.

The Cr parity plot shows wider scatter around the 45° line compared to Mo and Co. While the model tracked concentration ranges reasonably well, deviations above and below the diagonal indicate systematic prediction errors. The moderate R^2^ = 0.522 reflects this, with the model explaining about half of the observed variance. Predictions tended to underestimate at higher concentrations.

Chromium residuals show wider scatter, including underestimation at higher predicted values (negative residuals). The lack of random distribution suggests some bias in the model fit. Variability in residuals indicates reduced stability in prediction.

#### 3.7.3. Titanium

Titanium concentrations are well tracked by the RF, with predicted values closely following actual measurements, as seen in Figure 17. Small discrepancies appear in mid-level concentrations, but the overall temporal behavior is faithfully captured. This balance is reflected in a relatively high R^2^ of 0.707, with stable prediction performance across both low and high concentrations, supporting the model’s robustness.

For Ti, points are closely distributed along the diagonal, suggesting good predictive fidelity. A slight spread is visible at both low and high concentrations, but overall, the model effectively captured the concentration profile.

Titanium residuals are relatively balanced, distributed around zero with a slight negative skew at mid-to-high predicted values. While some bias exists, the overall spread is contained, supporting the model’s good performance (R^2^ = 0.707). Errors appear stable across the concentration range.

#### 3.7.4. Molybdenum

Molybdenum predictions approximate observed values across time but with smoothing of extremes. At higher concentrations (>6 µg/L), the model underpredicts, whereas mid-range predictions align more closely with measured values (Figure 18). The temporal trajectory is reasonably represented, confirming the Random Forest’s ability to capture nonlinear patterns. The corresponding R^2^ = 0.718 demonstrates that the model explains a substantial portion of the variance while missing some peak deviations.

The parity plot compares predicted versus observed Mo concentrations. Ideally, all points would align on the 45° diagonal, indicating perfect predictions. The model captured the general magnitude of concentrations but underestimated at higher actual values (>5 µg/L), as shown by points lying below the line. Statistically, the R^2^ value confirms that 71.8% of variance was explained, though residual deviations suggest smoothing of extremes.

Residuals for Mo fluctuate around zero but show a greater spread at higher predicted values. This mild difference suggests the model underestimated peak concentrations while overestimating some mid-range values. Nonetheless, residual symmetry supports moderate calibration.

#### 3.7.5. Nickel

For nickel, predictions follow the overall time-course of observed concentrations but with noticeable discrepancies. The model slightly underestimates higher values (>2 µg/L) and overestimates some lower values (Figure 19). The reduced data density contributes to variability, limiting predictive fidelity. Statistically, the R^2^ = 0.297 confirms weak explanatory power, indicating that nickel concentrations were less well represented by the model compared to other serums.

Nickel predictions are tightly clustered but deviate noticeably from the 45° line, reflecting systematic underprediction of actual concentrations. The parity plot confirms that the model did not generalize well for nickel data.

The new nickel residuals reveal consistent deviations above zero, reflecting systematic underestimation by the model. The lack of scatter variety indicates insufficient training data for nickel concentrations, reducing model generalizability. 

A summary of model performance metrics is highlighted in Table 10. 

## 4. Discussion

### 4.1. Contextualizing the Results

This work aimed to characterize the temporal patterns of metal ions in patients with hip prosthetics by using a pooled dataset across multiple biological fluids. The analysis was designed to capture both the early kinetic behavior and variability among long-term follow-up intervals by comparing compartmental differences between serum, whole blood, and urine. This study intended to integrate a machine learning model with observed data.

Most longitudinal studies report systemic Co and Cr levels in whole blood or serum, as they are most clinically relevant. There has not been universal agreement on the optimal fluid to analyze the systemic exposure of heavy metal ions, but each fluid can provide distinct insight into the clinical outcomes for a patient with a hip prosthetic device.

Cobalt, serum, and whole blood showed similar behavior in their models. The similarity between serum and whole blood cobalt reinforces the early rise and stabilization pattern across fluids. Variability bands were wider in serum during the early period, possibly reflecting a greater number of reported participants/cohorts at different postoperative intervals, whereas whole blood values clustered more narrowly around the mean.

Chromium demonstrated a different relationship between serum and whole blood. Unlike cobalt, serum and whole blood chromium diverged in magnitude, suggesting differences in distribution or clearance between compartments. As with cobalt, serum chromium values showed greater variability than whole blood, likely influenced by differences in cohort size and study heterogeneity at reported timepoints.

Titanium displayed a distinct rise–decay pattern compared with the persistent elevations of cobalt and chromium. This behavior is consistent with lower systemic persistence of titanium and suggests more effective clearance relative to cobalt and chromium. The trajectory highlights how alloy composition and corrosion mechanisms influence ion release and long-term systemic burden.

Machine learning analyses provided a complementary perspective, assessing how well temporal data predicted observed concentrations across ions. Predictive fidelity was highest for cobalt, with titanium and molybdenum also showing more similar performance. Chromium displayed only moderate accuracy, and nickel showed weak predictability, reflecting limited data across time and higher variability. These results emphasize that while regression models capture the average kinetic shape, the ability to forecast specific ion levels from time alone varies by element.

The pooled data highlight an early postoperative rise across ions, followed by stabilization or decline at different magnitudes depending on the element and fluid. Cobalt behaved similarly in serum and whole blood, while chromium revealed compartmental differences. Titanium diverged further, showing a rise–decay profile. These patterns, supported by machine learning performance metrics, demonstrate that pooled ion data yield reproducible trajectories and that variability constrains predictability.

### 4.2. Metal Ion Kinetic Synthesis

The modeled trajectories show that ions demonstrate similar early postoperative behaviors with different magnitudes. In essence, these results suggest that wide-scale ion kinetics follow a general course during the first three to four years postoperatively and are related to material and fluid compartmentalization properties. If concentrations deviate on the extreme ends of such concentrations; for example, constant Co elevation or different Cr levels across fluids may be indicative of a device-related problem or excessive tribocorrosion. Such patterns should be interpreted with clinical findings and device evaluation.

This analysis may provide a framework for what might be expected on a larger, population-level scale but should be interpreted cautiously. Further, multicenter prospective cohorts using standardized sampling should be conducted to confirm these trajectories and reduce confounding by implant design, analytical method, and patient-specific factors. If these patterns are supported, they may help surgeons detect early or late abnormal wear and corrosion. Knowledge of systemic exposure may also help anesthesiologists and pharmacists recognize potential drug–metal interactions or changes in drug behavior for patients with higher metal ion levels.

### 4.3. Mechanistic Basis for Compartmental Differences

The comparison of cobalt and chromium concentrations across serum, whole blood, and urine highlights the importance of compartmentalization in metal ion kinetics. Serum may represent the extracellular fraction where ions circulate immediately after release, whereas whole blood incorporates a substantial cellular volume that can dilute plasma concentrations when measured per unit volume. First, the hematocrit (about 45% of blood volume) reduces the apparent concentration of ions in whole blood compared with serum. Second, the main protein carriers for cobalt and chromium (albumin and transferrin [78,79]) are present in synovial fluid and confined to the plasma fraction, concentrating ions in serum rather than within cellular compartments. Third, ionic movement across erythrocyte membranes is limited, particularly for chromium in the trivalent state (Cr^3+^), which does not readily cross cell membranes [78]. Finally, methodological factors may contribute, since whole blood assays require cell lysis and digestion, while serum offers a simpler matrix that captures both free and protein-bound ions. Together, these can help explain why chromium appeared to be higher in serum than in whole blood in pooled models, while cobalt, because of its greater potential for erythrocyte uptake, showed comparable levels across both fluids.

At the molecular level, cobalt and chromium differ in their protein binding and distribution, which can help explain why cobalt typically appears to be higher than chromium in whole blood. Cobalt is predominantly present as Co^2+^, likely showing a relatively weak affinity for transferrin compared with trivalent chromium and iron. Much of the circulating cobalt is bound to albumin, while a measurable fraction remains unbound and readily filterable [72,73]. Erythrocyte partitioning likely offsets the dilutional effect of hematocrit, resulting in similar concentrations of cobalt in serum and whole blood. Some Co partitions into RBCs because of its binding with hemoglobin, but the extent of this association in vivo is uncertain [80].

By contrast, chromium circulates primarily as Cr^3+^, which binds tightly to transferrin and other plasma proteins [77,78]. Because this protein-bound fraction does not readily cross red cell membranes, chromium may remain underrepresented in whole blood despite being present at similar levels to cobalt in serum. Cr^3+^ binds tightly to transferrin and remains confined to the plasma fraction, whereas Cr^6+^, which is more membrane-permeable, would be expected to enter erythrocytes and yield higher whole blood levels. Although our analysis cannot directly determine oxidation state, the serum–whole blood difference supports the interpretation that most chromium released from implants is present in the trivalent form.

These compartmental dynamics are further shaped by clearance mechanisms. Cobalt’s weaker binding may allow for greater systemic mobility and rapid renal excretion, which is evident in results where average urinary Co exceeds Cr. In the pooled dataset, urinary Co often exceeded urinary Cr, especially in the middle and late phase follow-ups. This can reflect a larger freely filterable fraction for Co and chromium’s tendency to bind with plasma proteins, making it less filterable.

From a monitoring perspective, this compartmental framework helps explain why some researchers prioritize whole blood for long-term surveillance, as it provides more reproducible measurements across laboratories and is less influenced by pre-analytical variability. Serum and urine, however, may be more sensitive to short-term spikes or clearance dynamics. For researchers, this underscores that serum, whole blood, and urine are not interchangeable measures of exposure but complementary markers reflecting different aspects of metal kinetics.

Figure 20 shows a proposed behavior for Co and Cr after implant release. After they enter the vasculature, they can partition among serum proteins, blood cell compartments, or a freely filterable fraction. This broad proposed pathway may help explain how patterns are quantified. First, Co shows similar behavior in serum and whole blood, consistent with extracellular surge followed by cellular uptake and efficient filtration. Second, Cr stabilizes higher in serum than in whole blood, which aligns with protein binding with limited erythrocyte permeability. Third, Co clearance through urine rises across its time phase more than chromium, which can reflect a larger filterable fraction.

In the pooled dataset, long-term means usually fell within 1–3 μg/L across fluids. Biological responses to wear and corrosion are on a spectrum, rather than an all-or-nothing switch. Figure 21 demonstrates this as a conditional pathway. Early ion and particle release can trigger low-level innate signaling, cytokine production, and routine bone remodeling.

The fluid patterns can infer how ions move and where they reside. The next question is how the surrounding tissue responds to such exposure. Figure 21 shows a conditional pathway beginning with ions entering the periprosthetic space and potential downstream effects.

### 4.4. Empirical Evidence from Literature

Across reports, Co and Cr concentrations have been described as higher in serum than in whole blood. Daniel et al. demonstrated this difference, showing higher Co and Cr values in serum and emphasizing that the two matrices should not be used interchangeably [81]. Malek et al. supported this, noting that attempts to apply conversion factors between serum and whole blood are often unreliable because of concentration-dependent variability [82]. Smolders et al. derived predictive equations to estimate whole blood values from serum, with typical errors within ±1.0 μg/L, but still cautioned that the two fluids cannot be considered interchangeable [83].

Walter et al. analyzed the distribution of Co and Cr across whole blood, plasma, serum, and erythrocytes and reported that the majority of ions were localized in the extracellular compartments. Concentrations were highest in serum and plasma, with the lowest levels observed in red blood cells, leading the authors to recommend serum or plasma as the preferred monitoring fluid for systemic ion levels [80]. A systematic review of 16 different MoM implant types reinforced this pattern but also highlighted variability, reporting Cr values between 0.5 and 2.5 μg/L in blood and 0.8 and 5.1 μg/L in serum, while Co ranged from 0.7 to 3.4 μg/L in blood and 0.3 to 7.5 μg/L in serum [84]. The wider variability in serum could reflect its sensitivity to implant performance and clearance patterns, whereas whole blood values may be buffered by the ion content in cellular components, producing narrower ranges. Our pooled findings mirror this heterogeneity: Cr concentrations were higher in serum than in whole blood, consistent with prior reports, whereas cobalt values were more comparable between fluids.

For other ions, the literature and our pooled dataset both indicate limited systemic representation. Nickel concentrations were consistently low across serum, whole blood, and urine relative to cobalt and chromium [10]. The association between titanium ion levels and implant performance remains unclear, and threshold values for different implant designs have not been established. Recent attempts analyzing blood titanium levels as a biomarker for implant function showed that patients with “massive acetabular implants” had significantly higher Ti levels than patients with “standard hip implants” [85]. Molybdenum is rarely monitored, but experimental work has shown its dissolution and binding to albumin [86], while clinical studies emphasize its efficient renal clearance [87].

Together, the literature reinforces two key features that emerged in our analysis: (1) serum Cr exceeds whole blood, likely reflecting its strong protein binding and extracellular confinement, and (2) cobalt displays more comparable concentrations between serum and whole blood, plausibly due to cellular uptake that diminishes the serum–blood difference.

### 4.5. Machine Learning Models

The application of Random Forest (RF) regression to the study-level serum ion dataset provided a complementary perspective to the pooled nonlinear regression models. Whereas exponential association fits summarized average trajectories, the RF approach tested whether predictive algorithms could capture temporal concentration patterns across the full range of reported values.

Model performance varied substantially across ions. For cobalt, RF predictions closely tracked measured values (R^2^ = 0.861), reproducing both the early postoperative rise and subsequent stabilization with only minor underestimation at peak values. This high fidelity reflects the relative consistency of Co distributions across studies, enabling the model to capture central tendencies reliably. Ti and Mo also showed moderate predictive accuracy (R^2^ = 0.707 and 0.718, respectively), suggesting that their temporal patterns were reproducible despite smaller sample sizes.

By contrast, Cr was more challenging to model (R^2^ = 0.522). The algorithm successfully captured overall temporal trends but systematically underestimated higher concentrations, leading to wider scatter in the parity plots. This may reflect the biological and analytical variability of chromium in its trivalent state, binding affinity, cellular interactions, and renal clearance. Nickel could not be modeled reliably (R^2^ = 0.297), a result consistent with its sparse representation in the literature and limited data.

Taken together, the machine learning results highlight the complementary role of RF relative to nonlinear regression. While pooled exponential models provide interpretable averages that illustrate kinetic shapes, they are sensitive to late-stage data and influenced by single elevated values. RF accommodates nonlinearities and variance across the full dataset, offering more predictions given and interstudy differences.

Limitations should also be acknowledged. Because data was combined from study-level values, and there were fewer participants in the late follow-up periods, the model could be constrained. Since open-access data was used, the RF models are not intended to be used as a clinical prediction. Rather, the goal was to explore its methodological potential.

### 4.6. Toxicity Thresholds and Reference Ranges

The clinical relevance of metal ion kinetics depends on how observed concentrations compare to proposed thresholds and reference ranges. Proposed thresholds and risk limits for Co and Cr are summarized in Table 11 and Table 12 from regulatory agencies and peer-reviewed studies. Hart et al. claimed that the 7 ppb cutoff value proposed by the MHRA had a specificity of 89% and sensitivity of 52.%. Alternatively, their proposed cutoff of 4.97 ppb had a slightly lower specificity (86%) but higher sensitivity (63%).

With the growing use of titanium-based alloys in THA, there is an increasing need for clinical monitoring and the establishment of clear thresholds or risk profiles. One recent study proposed threshold values of 2.20 μg/L in blood and 2.56 μg/L in plasma for patients with well-functioning titanium hip implants at mid- to long-term follow-up [79].

Two clinical laboratories’ reference ranges for Co, Cr, Ti, Mo, and Ni in whole blood and serum are summarized in Table 12. These concentrations represent baseline concentrations in healthy adults and provide context for interpreting post-implant ion levels.

### 4.7. Elevated Ion Outlier Analysis

Elevated Co and Cr are implicated in ALTR, such as necrosis, pseudotumors, and ALVAL-type lesions. Bradberry et al. reviewed published cases of systemic cobalt toxicity and identified three major patterns of involvement: neuro-ocular, cardiac, and thyroid [93].

Crutsen et al. reviewed 67 case reports encompassing 79 patients with markedly elevated Co concentrations, a condition referred to as cobaltism. Elevated levels were consistently present at the onset of systemic symptoms, which most frequently involved neurological and cardiovascular manifestations. Among reported cases, 24% of symptoms were sensory in nature, 19.3% involved the central or peripheral nervous system, and 22.1% were attributed to cardiovascular complications [19].

Other studies showed that elevated blood cobalt levels could predict the formation of pseudotumors without the effect of Cr [94]. However, the cobalt-to-chromium ratio itself has not been shown to be a reliable biomarker for predicting ALTRs, as variability in alloy composition, ion solubility, and renal clearance limits its clinical use [95].

Table 13 identifies timepoints where cobalt or chromium concentrations exceeded the 7 µg/L threshold proposed by the MHRA. Threshold exceedance was common within the first 1–3 years postoperatively, which aligns with the “running-in” phase of wear. This is when surfaces undergo early adaptation and passive oxide films are disrupted. Later outliers (e.g., 72–108) may suggest that elevated values can also be related to corrosion at tapers or tribocorrosion in the long run.

Not all studies followed patients beyond 60–100 months, and not all documented whether threshold exceedance was linked to clinical symptoms. As a result, outliers should not be interpreted as definitive markers of implant failure. Instead, they reflect the differences in ion release across devices and patients, influenced by factors such as design, patient-specific factors, and analytical techniques.

In the pooled analysis, most concentrations remained below regulatory thresholds, and long-term averages were found at levels between 1 and 3 µg/L. In other words, threshold exposure is not the typical trajectory among cohorts with hip implants, and common ion exposure is below the threshold.

### 4.8. Limitations

This study is subject to limitations. The analysis relied on study-level summary statistics rather than raw patient-level data, which was unavailable. This reduced the ability to model true individual trajectories. There was also diversity across the included studies in implant design, sample size, analytical assay techniques, patient demographics, and follow-up intervals, which most likely contributed to the degree of spread. This made it more difficult to reduce confounding. Selection bias may also be present, as easily accessible data was used, which led to underrepresentation of certain patient populations and implant designs. Data collection was uneven across each of the metals, with Co and Cr being reported more than Ti, Mo, or Ni. Not all studies reported individual variance, which means that SD was derived from inter-study variability. Finally, the machine learning models were intended to be illustrative and exploratory. Work in the future should focus on standardized sampling across multiple biological fluids while incorporating implant and patient-specific factors into their models. If these gaps are addressed, future work can shift to risk stratification and help improve surveillance of patients with hip implants.

## 5. Conclusions

This study compiled data on serum, whole blood, and urine metal ion concentrations after different follow-up times from cohorts with hip implants. It integrated a pooled analysis with exploratory kinetic models, including nonlinear regressions and machine learning. Cobalt and chromium demonstrated consistent long-term elevations with moderate increases dependent on time. While titanium showed minimal accumulation. Nickel and molybdenum were characterized by high variability and limited reporting. In some cases, machine learning models suggest that cohort variability can distort average trends, which can emphasize the need for individualized monitoring strategies. While time contributes to rising concentrations, it appears to mainly influence early rises, which indicates that other patient- and implant-specific factors are also influential in the long run. Our findings highlight the value of fluid-specific reference kinetics for postoperative monitoring, suggesting that serum Co and Cr provide complementary indicators of implant wear and systemic exposure.

## Figures and Tables

**Figure 1 bioengineering-13-00044-f001:**
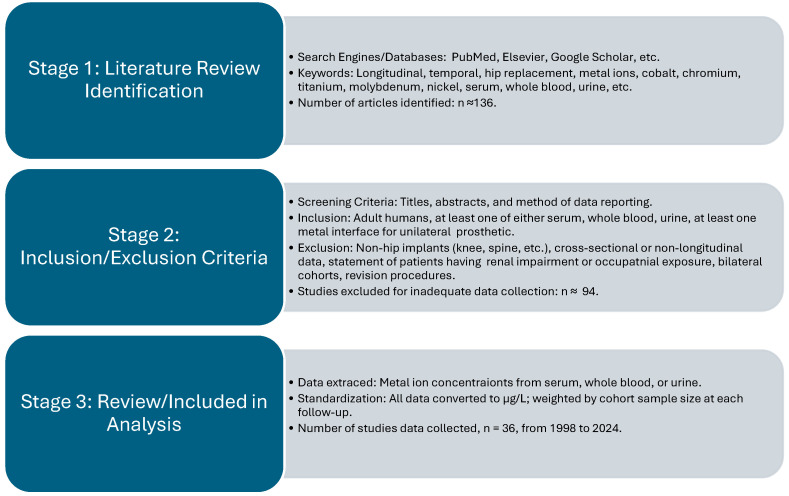
Schematic summarizing literature search, screening, and inclusion process for studies that report longitudinal metal ion levels in patients with hip prosthetics. The search was performed using different databases and search engines with keywords reflecting longitudinal metal ion concentrations. Thirty-six studies met the inclusion criteria and were utilized for data extraction in the pooled analysis.

**Figure 2 bioengineering-13-00044-f002:**
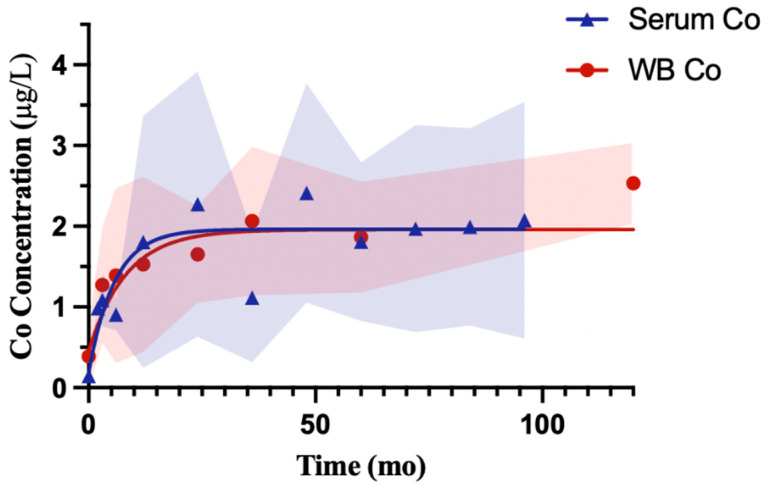
Modeled trajectories of serum and whole blood Co concentrations. Pooled and weighted values were modeled using a one-phase exponential association. Serum (blue) rose from 0.18 µg/L to a plateau of 1.96 µg/L with a half-time of 3.9 months $\left[Y\left(t\right)=0.18+1.8(1-e^{-0.18\left(t\right)})\right]$, while whole blood (red) rose from 0.45 µg/L to a plateau of 1.96 µg/L with a halftime of 5.7 months $\left[Y\left(t\right)=0.45+1.5(1-e^{-0.12\left(t\right)})\right]$. Shaded regions represent the standard deviation (SD) of mean values at each follow-up month.

**Figure 3 bioengineering-13-00044-f003:**
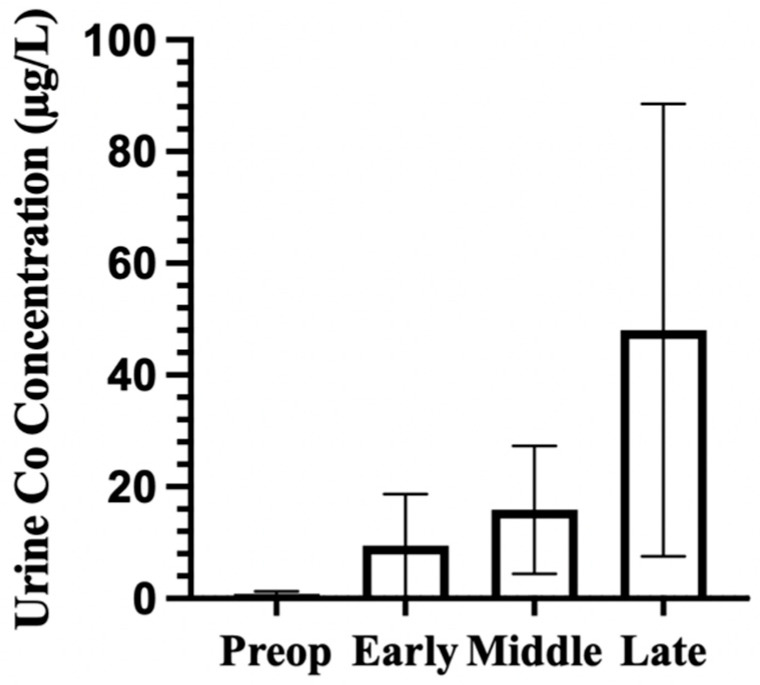
Urinary cobalt concentrations across time phases. Bars represent mean urinary Co concentrations across each phase. Error bars represent standard deviation (SD). Time phases were defined as preop (before implantation), early (1–23 months), middle (24–47 months), and late (≥48 months). Mean urinary Co tended to increase across time phases, with greater variability in the late phase.

**Figure 4 bioengineering-13-00044-f004:**
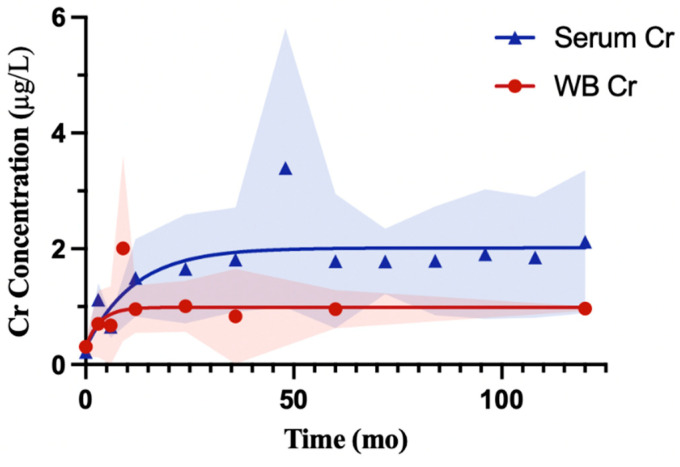
Modeled trajectories of serum and whole blood Cr concentrations. Pooled and weighted values were modeled using a one-phase exponential association. Serum (blue) rose from 0.28 µg/L to a plateau of 2.02 µg/L with a half-time of 8.0 months $\left[Y\left(t\right)=0.28+1.7(1-e^{-0.09\left(t\right)})\right]$. Whole blood (red) rose from 0.3 µg/L to a plateau of 0.99 µg/L with a half-time of 2.4 months $\left[Y\left(t\right)=0.30+0.69(1-e^{-0.29\left(t\right)})\right]$. Shaded regions represent the standard deviation (SD) of pooled values at their respective follow-ups.

**Figure 5 bioengineering-13-00044-f005:**
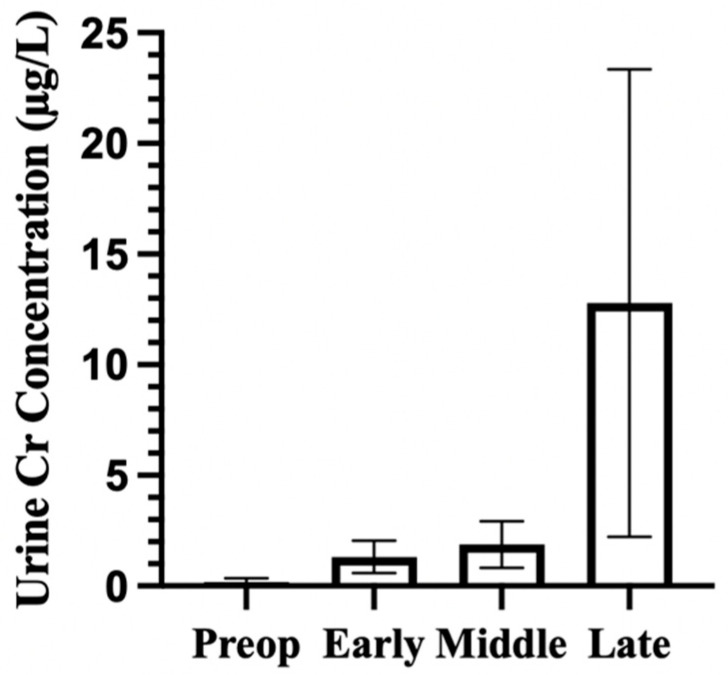
Urinary chromium concentrations across pooled time phase following hip implantation. Pooled mean urinary chromium concentrations (µg/L) are shown across four time phases: preoperative, early (1–23 months), middle (24–47 months), and late (≥48 months). Bars represent mean values, with error bars indicating standard deviation (SD). Values reflect descriptive pooling across studies that were included.

**Figure 6 bioengineering-13-00044-f006:**
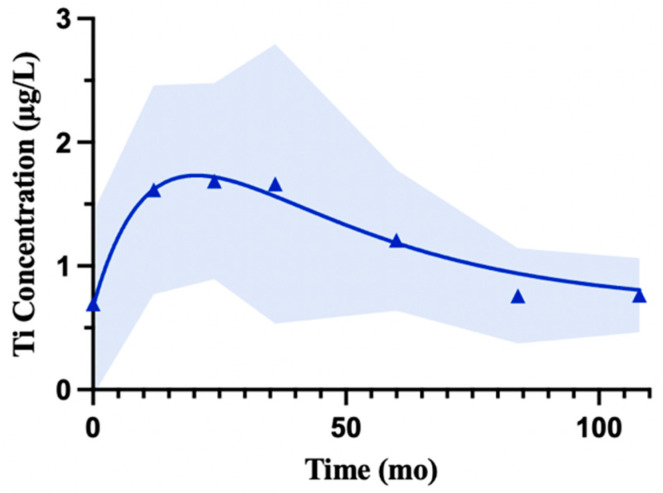
Modeled trajectory of serum Ti concentrations. Pooled and weighted serum Ti concentrations were captured using an exponential rise-then-decay association. Concentrations rose from a baseline of 0.67 µg/L to a modeled peak around 12–24 months, followed by a gradual decline toward lower levels over time. The fitted function was $[Y\left(t\right)=0.67+3.0(e^{-0.028t}-e^{-0.049t})]\;$ with best-fit rate constants of k = 0.049 and k_d_ = 0.028. Shaded regions represent the standard deviation at each follow-up month.

**Figure 7 bioengineering-13-00044-f007:**
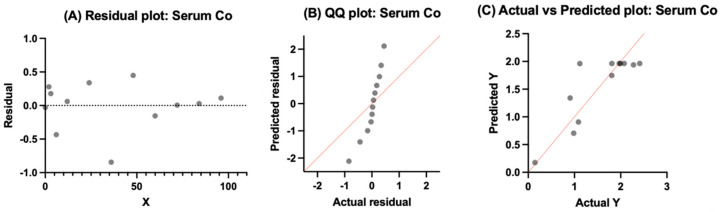
Model output for serum Co using a one-phase exponential association. (**A**) Plot of residuals against time does not reveal strong systematic bias. (**B**) QQ plot showing deviation from normality and curvature away from the angled line. (**C**) Actual vs. predicted plot values fall close to the identity line with not much scatter, consistent with adequacy of fit in the pooled data.

**Figure 8 bioengineering-13-00044-f008:**
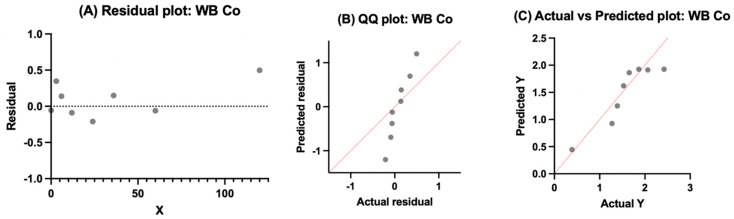
Model output for whole blood Co modeled with a one-phase exponential association. (**A**) Residuals plotted against time showing no systemic trends. (**B**) QQ plot demonstrating approximate normality of residuals, with deviations at the tails showing non-normal distribution for residuals. (**C**) The actual vs. predicted points are clustered near the identity line without great scatter, pointing toward adequacy of model fit in whole blood data.

**Figure 9 bioengineering-13-00044-f009:**
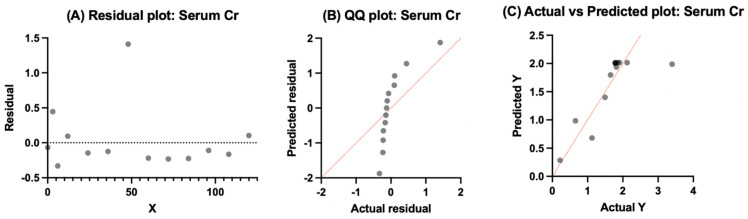
Model output for serum chromium modeled with a one-phase exponential association. (**A**) Residuals plotted against time are centered near zero without strong systematic bias, though a few outliers are present. (**B**) QQ plot showing curvature away from the identity line, indicating deviations from normality. (**C**) Actual vs. predicted values generally cluster near the identity line with moderate scatter, with one point outside the expected range, consistent with adequacy of the model fit for pooled serum chromium data.

**Figure 10 bioengineering-13-00044-f010:**
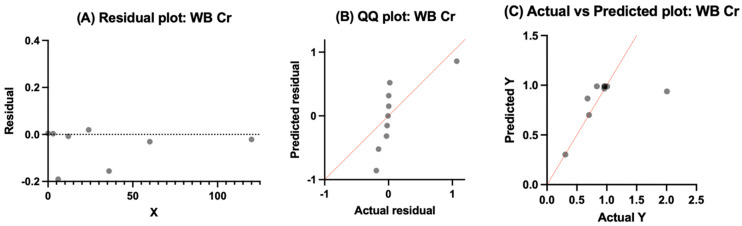
Model output for whole blood chromium modeled with a one-phase exponential association. (**A**) Residuals plotted against time are centered around zero without systematic bias. (**B**) QQ plot showing approximate normality with modest deviations at distribution tails. (**C**) Actual vs. predicted values closely cluster along the identity line, with one point outside the expected range, consistent with adequacy of model fit for pooled whole blood chromium data.

**Figure 11 bioengineering-13-00044-f011:**
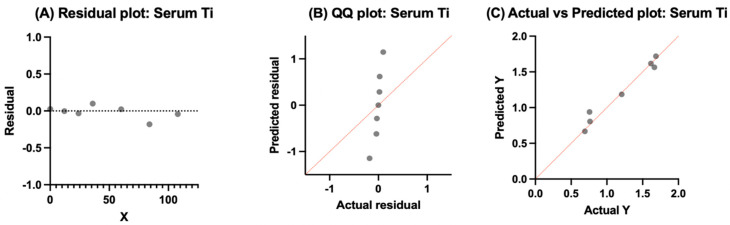
Model output for serum titanium using an exponential rise–decay association. (**A**) Residuals plotted against time are centered around zero without much evidence of systematic bias. (**B**) QQ plot showing a near-vertical distribution of points, indicating narrow variance relative to normal expectations. (**C**) Actual vs. predicted values cluster tightly along the identity line without extreme scatter, supporting an adequate model.

**Figure 12 bioengineering-13-00044-f012:**
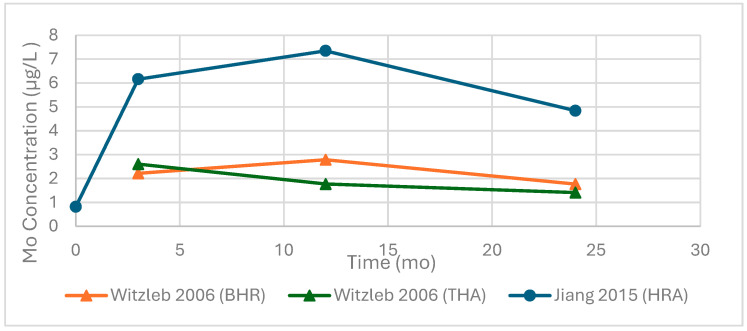
Reported serum molybdenum (Mo) concentrations over time following primary hip arthroplasty. Values were extracted directly from individual studies. Circular markers denote reported means, while triangular markers denote reported medians. All concentrations are in micrograms per liter (µg/L). Data include total hip arthroplasty (THA), Birmingham Hip Resurfacing (BHR), and hip resurfacing arthroplasty (HRA), as indicated by the study. Data adapted from [39,75].

**Figure 13 bioengineering-13-00044-f013:**
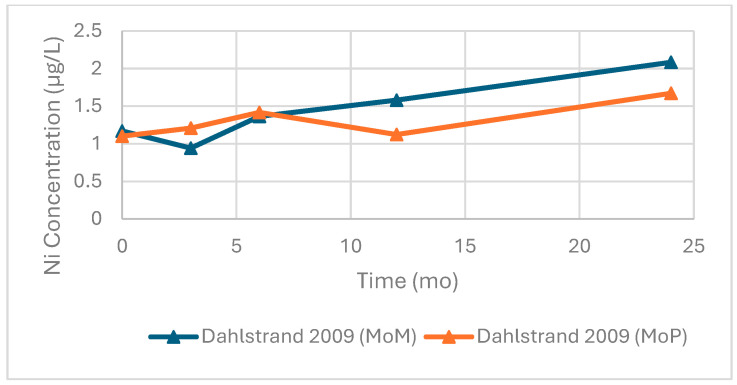
Serum nickel (Ni) concentrations following primary total hip arthroplasty as seen from Dahlstrand et al. (2009) [38]. Data are shown for patients with metal-on-metal (MoM, n = 28) and metal-on-polyethylene (MoP, n = 26) prostheses at preop and 6, 12, and 24 months after surgery. In both groups, Ni concentrations increased over time, with the MoM cohort showing a slightly greater overall rise.

**Figure 14 bioengineering-13-00044-f014:**
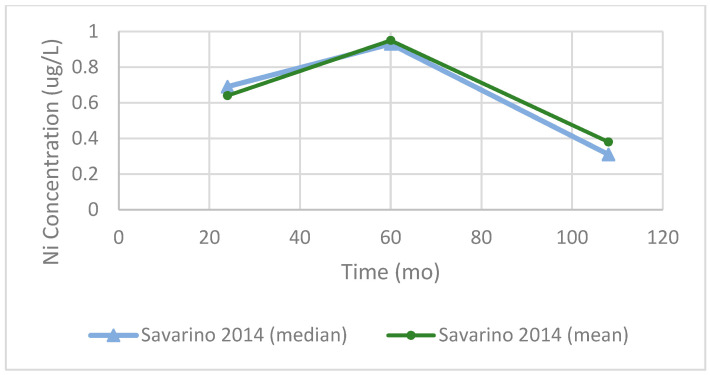
Serum Ni concentrations after Birmingham Hip Resurfacing (BHR). Data from Savarino et al. (2014) [77] are shown as both reported means (green circle) and medians (blue triangle) at 24, 60, and 108 months postoperatively. Concentrations peaked at 60 months before declining at longer-term follow-up.

**Figure 15 bioengineering-13-00044-f015:**
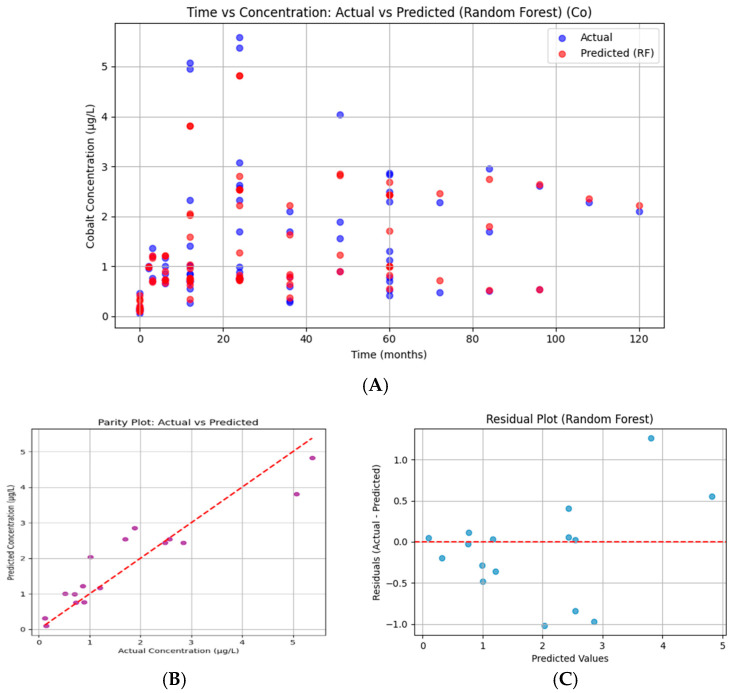
Random Forest regression for serum cobalt. (**A**) Time vs. concentration showing actual values (blue) and model predictions (red). (**B**) Parity plot of predicted versus observed concentrations, with points closely aligned to the 45° diagonal. (**C**) Residuals plotted against predicted values are distributed symmetrically around zero. Model performance: MAE = 0.417, RMSE = 0.573, R^2^ = 0.861.

**Figure 16 bioengineering-13-00044-f016:**
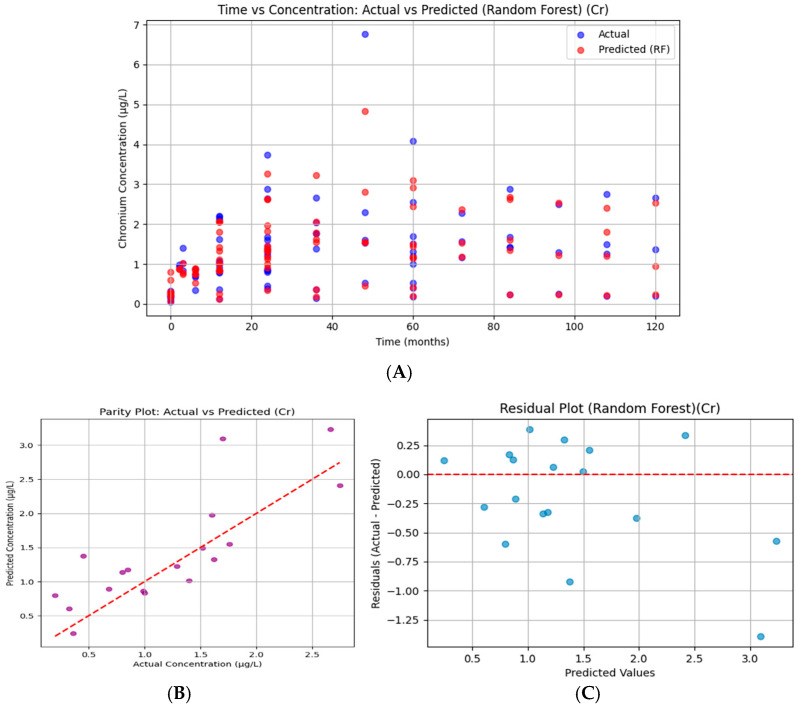
Random Forest regression for serum chromium. (**A**) Time versus concentration showing observed values (blue) and Random Forest predictions (red). Predictions captured overall patterns but underestimated high peaks and overestimated mid-range concentrations. (**B**) Parity plot of predicted versus observed concentrations, showing wider scatter around the 45° line and indicating systematic deviations (R^2^ = 0.522, RMSE = 0.495). (**C**) Residual plot demonstrating greater variability and bias at higher predicted values, reflecting moderate predictive accuracy and reduced stability relative to cobalt.

**Figure 17 bioengineering-13-00044-f017:**
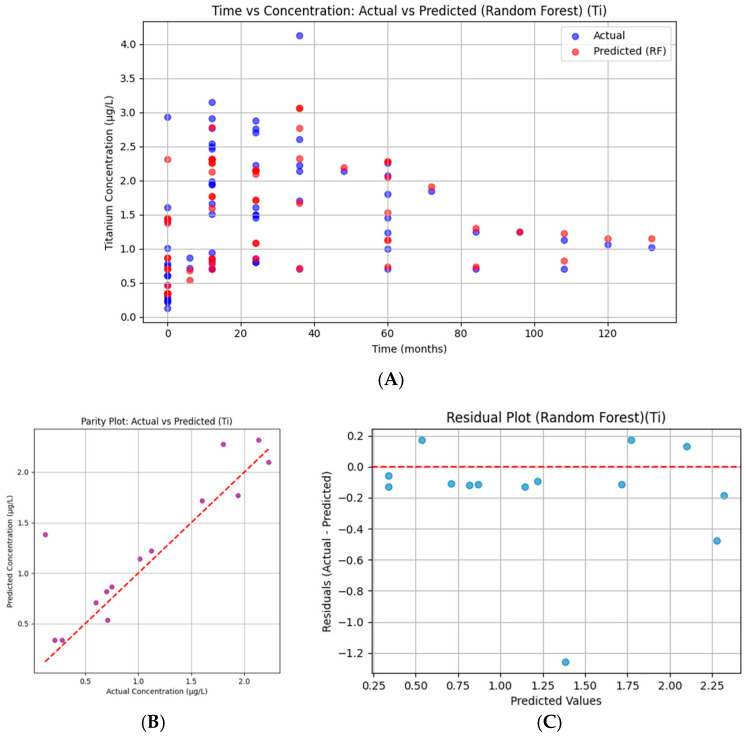
Random Forest regression for serum titanium. (**A**) Time versus concentration showing observed values (blue) and Random Forest predictions (red). Predicted values tracked actual concentrations closely, with minor deviations at mid-levels. (**B**) Parity plot of predicted versus actual concentrations, demonstrating close alignment with the 45° diagonal and confirming robust explanatory power (R^2^ = 0.707, RMSE = 0.380). (**C**) Residual plot showing values distributed around zero with slight negative skew at higher predicted levels, supporting stable and reliable model performance.

**Figure 18 bioengineering-13-00044-f018:**
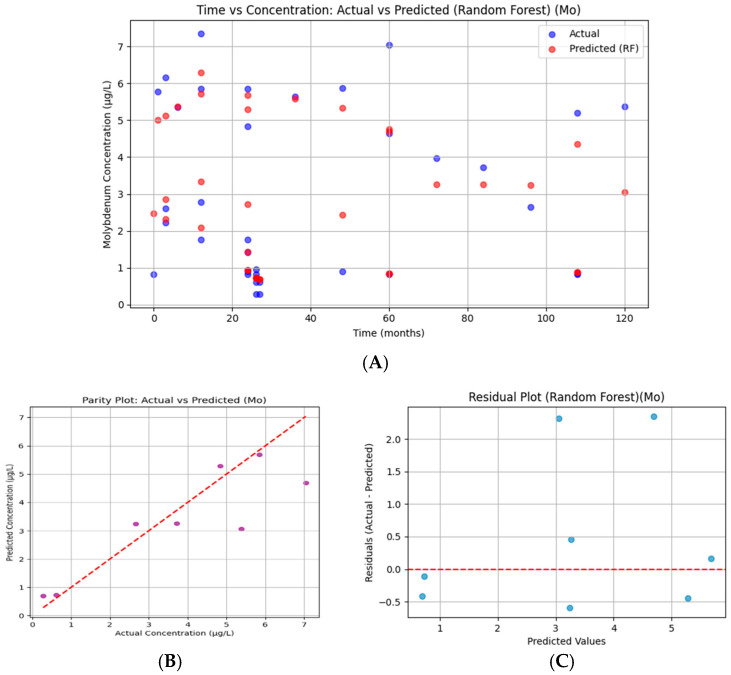
Random Forest regression for serum molybdenum. (**A**) Time versus concentration showing observed values (blue) and Random Forest predictions (red). Predictions reproduced the temporal trajectory but smoothed concentration extremes, with underestimation at higher values. (**B**) Parity plot of predicted versus actual concentrations, with most points aligned near the 45° diagonal but showing underprediction at higher actual values (>5 µg/L). Model performance: R^2^ = 0.718, RMSE = 1.218. (**C**) Residual plot demonstrating values centered near zero with wider spread at higher predictions, indicating mild heteroscedasticity but overall stable model calibration.

**Figure 19 bioengineering-13-00044-f019:**
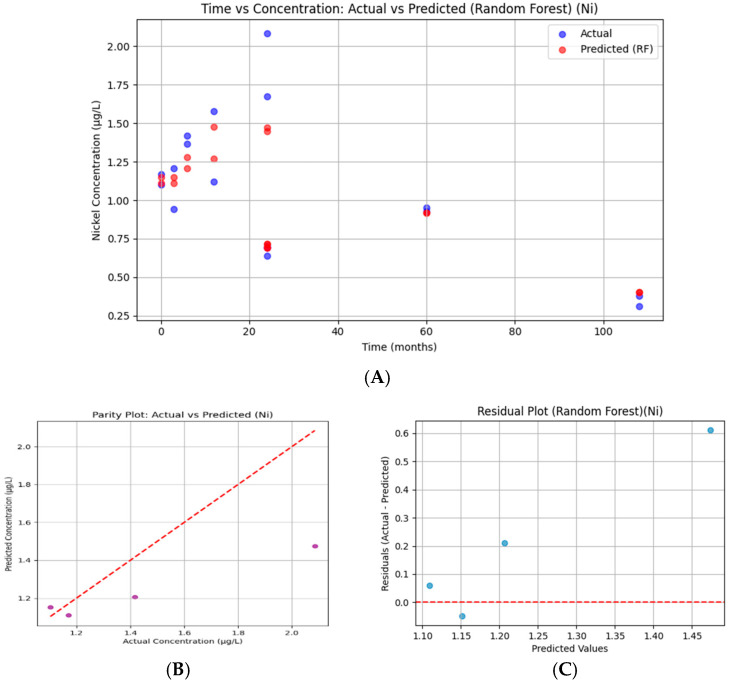
Random Forest regression for serum nickel. (**A**) Time versus concentration showing observed values (blue) and Random Forest predictions (red). Predictions approximated overall temporal patterns but systematically underestimated higher concentrations. (**B**) Parity plot of predicted versus actual values showing deviations from the 45° diagonal, consistent with limited predictive fidelity (R^2^ = 0.297, RMSE = 0.325). (**C**) Residual plot indicating consistently positive errors, reflecting systematic underestimation of nickel values. Reduced data density limited the model’s explanatory capacity compared with other ions.

**Figure 20 bioengineering-13-00044-f020:**
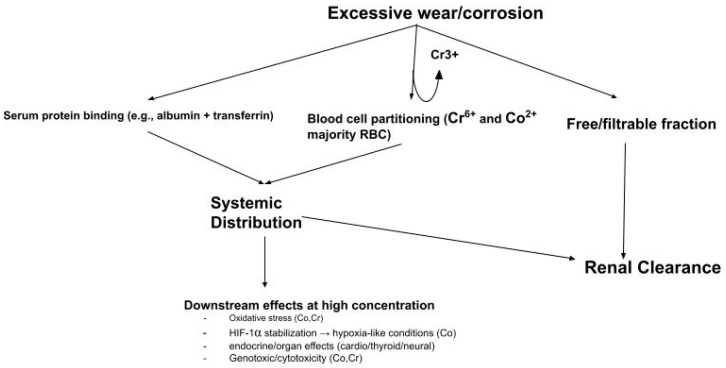
Proposed systemic distribution, binding, and clearance of metal ions. Excessive wear and corrosion create ions that partition among serum proteins, blood cell components, and filterable fractions, which can be renally excreted. If present at high enough concentrations, downstream effects may occur.

**Figure 21 bioengineering-13-00044-f021:**
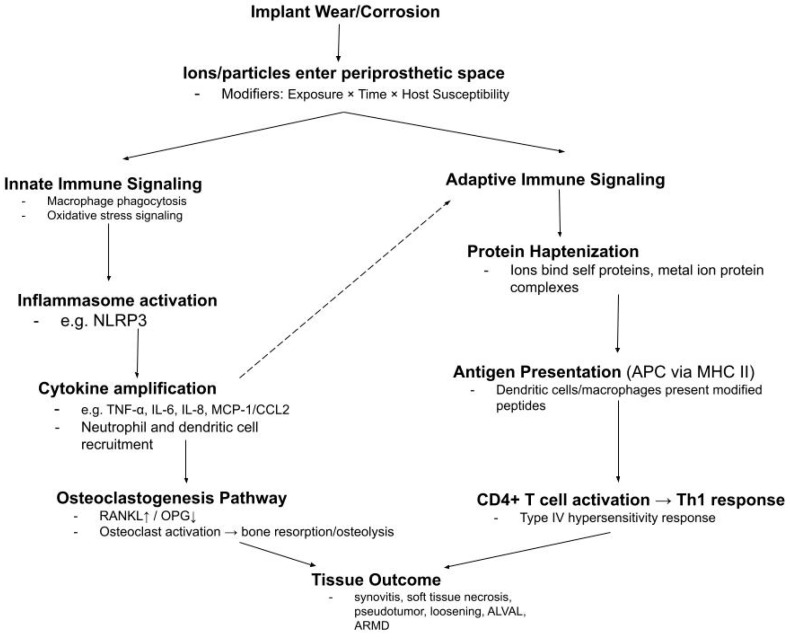
Conditional implant-to-immune pathway after metal ion exposure. Wear and corrosion create ions and particles that can enter joint space and trigger innate and adaptive responses. Progression varies with exposure, time, and host susceptibility and provides context for subthreshold kinetics observed but also for occasional outlier elevation.

**Table 1 bioengineering-13-00044-t001:** Nominal chemical compositions of common orthopedic implant alloys standardized by ASTM and ISO specifications. Major alloying elements are shown as ranges in weight percent, with trace elements included where applicable. “≤” refers to the maximum amount of corresponding metal. “Bal.” indicates balance, meaning the remainder of the composition is made up of the listed base element to reach 100%.

Alloy/Group	ASTM	ISO	UNS	Co	Cr	Mo	Ti	Ni	Al	V	Fe	Si	W	Mn	Nb
Cast CoCrMo	F75	5832-4	R30075	Bal.	27–30	5–7	≤0.1	≤0.5	≤0.1	-	≤0.75	≤1	≤0.2	≤1	-
Wrought CoCrMo	F1537	5832-12	R31538	Bal.	26–30	5–7	-	≤1	-	-	≤0.75	≤1	-	≤1	-
Wrought CoCrWNi	F90	5832-5	R30605	Bal.	19–21	-	-	9–11	-	-	≤3	≤0.4	14–16	1–2	-
Forged CoCrMo	F799	5832-12	R31538	Bal.	26–30	5–7	-	≤1	0.3–1	-	≤0.75	≤1	-	≤1	-
Wrought CoNiCr	F562	5832-6	R30035	Bal.	19–21	9–10.5	≤1	33–37	-	-	≤1	≤0.15	-	≤0.15	-
Wrought TiAlV	F136	5832-3	R56401	-	-	-	Bal.	-	5.5–6.5	3.5–4.5	≤0.25	-	-	-	-
Wrought TiAlNb	F1295	5832-11	R56700	-	-	-	Bal.	-	5.5–6.5	-	-	≤0.25	-	-	6.5–7.5
Wrought SS 316L	F138	5832-1	S31673	-	17–19	2.25–3	-	13–15	-	-	Bal.	≤0.75	-	≤2	-
Wrought SS 316L	F139	5832-1	S31673	<0.1	17–19	2.25–3	-	13–15	-	-	Bal.	≤0.75	-	≤2	-
Cast TiALV	F1108		R56406	-	-	-	Bal.	-	5.5–6.75	3.5–4.5	≤0.3	-	-	-	-
Wrought CrNiMo SS	F1350		S31673	-	17–19	2.25–3	-	13–15	-	-	Bal.	≤0.75	-	≤2	-
Wrought TiAlV	F1472	5832-3	R56400	-	-	-	Bal.	-	5.5–6.75	3.5–4.5	≤0.33	-	-	-	-
Wrought CoCrFeNiMo	F1058	5832-7	R30003	39-41	19–21	6–8	Bal.	14–16	-	-	Bal.	≤1.2	-	1.5–2.5	-
Wrought CoCrFeNiMo	F1058	5832-7	R30008	39-42	18.5–21.5	6.5–7.5	Bal.	15–18	-	-	Bal.	≤1.2	-	1–2	-
cp Ti	F67	5832-2	R50700	-	-	-	Bal.	-	-	-	≤0.5	-	-	-	-
Wrought CrNiMnMo SS	F1586	5832-9	S31675	-	19.5–22	2–3	-	9–11	-	-	Bal.	≤0.75	-	2–4.25	0.25–0.8

**Table 2 bioengineering-13-00044-t002:** Data coverage for pooled serum (blue) and whole blood (red) cobalt concentrations shown in Figure 2.

	Serum		Whole Blood	
Time (Months)	Reported Participants	Contributing Cohorts	Reported Participants	Contributing Cohorts
0	428	10	452	13
2	112	2	-	-
3	213	4	95	3
6	260	6	194	6
12	636	14	324	11
24	711	15	344	10
36	294	6	144	3
48	190	4	-	-
60	471	10	266	8
72	69	2	-	-
84	161	3	-	-
120	-	-	124	2

**Table 3 bioengineering-13-00044-t003:** Cohorts and participants contributing to pooled urinary cobalt concentration phase.

Time Phase	Reported Participants	Contributing Cohorts
Preop	103	3
Early	155	5
Middle	139	5
Late	103	3

**Table 5 bioengineering-13-00044-t005:** Cohorts and participants contributing to pooled urinary chromium concentration bins.

Time Phase	Reported Participants	Contributing Cohorts
Preop	103	3
Early	155	5
Middle	139	5
Late	103	3

**Table 6 bioengineering-13-00044-t006:** Data coverage for pooled weighted serum titanium concentrations shown in Figure 6.

Time (Months)	Reported Participants	Contributing Cohorts
0	388	15
12	412	16
24	268	11
36	161	6
60	200	7
84	94	2
108	66	2

**Table 7 bioengineering-13-00044-t007:** One-phase association model output for serum and whole blood Co and Cr. Best-fit values, 95% confidence intervals, and goodness-of-fit statistics are shown for serum and whole blood Co and Cr trajectories modeled with a one-phase exponential association. Variables include baseline concentration (*Y*_0_), plateau concentration, rate constant (*K*), time constant (*τ = 1/K*), half-time (*t_½_ = ln2/K*), and span (*plateau—Y*_0_). Goodness-of-fit statistics (R^2^, coefficient of determination; Sy.x, standard deviation of residuals, sum of squares; df, degrees of freedom) are provided.

One Phase Association	Serum Co	Whole Blood Co	Serum Cr	Whole Blood Cr
Best-fit values				
Y0	0.175	0.454	0.285	0.304
Plateau	1.964	1.958	2.015	0.989
K	0.176	0.121	0.086	0.288
Tau	5.688	8.275	11.57	3.471
Half-time	3.943	5.736	8.021	2.406
Span	1.789	1.504	1.731	0.685
95% CI (profile likelihood)				
Y0	0.067 to 0.283	0.378 to 0.530	0.185 to 0.382	0.254 to 0.353
Plateau	1.91 to 2.02	1.885 to 2.038	1.961 to 2.071	0.957 to 1.021
K	0.152 to 0.205	0.095 to 0.156	0.076 to 0.099	0.228 to 0.376
Tau	4.878 to 5.598	6.413 to 10.57	10.09 to 13.24	2.685 to 4.382
Half-time	3.381 to 4.573	4.445 to 7.328	6.995 to 9.179	1.843 to 3.038
Goodness of Fit				
df	3619	1940	3688	1936
R-squared	0.197	0.364	0.218	0.213
Sum of Squares	5423	1203	4153	562.2
Sy.x	1.224	0.788	1.061	0.539

**Table 8 bioengineering-13-00044-t008:** Exponential rise-and-decay output for serum Ti. Best-fit values and goodness-of-fit statistics are shown for serum titanium trajectories modeled with an exponential rise–decay association. Variables include baseline concentration (Y_0_), scaling parameter (A), and rate constants for the rising (kᵣ) and decaying (k_d_) phases. Goodness-of-fit statistics (R^2^, coefficient of determination; Sy.x, standard deviation of residuals; degrees of freedom; sum of squares) are provided.

Exponential Rise then Decay	Serum Ti
Best-fit values	
Y0	0.6668
A	~3.000
kr	0.04915
kd	0.02836
Goodness of Fit	
Degrees of Freedom	1520
R-squared	0.2215
Sum of Squares	932.0
Sy.x	0.7830

**Table 9 bioengineering-13-00044-t009:** Summary of cross-sectional reported mean serum molybdenum (Mo) concentrations following primary MoM total hip arthroplasty (THA) or Birmingham Hip Resurfacing (BHR). DL = detection limit.

Study	Fluid	Average Follow-Up (Months)	Mean Mo Concentration (µg/L)	Hip Device	Number of Patients
Moroni 2010 [76]	Serum	58	0.84	MoM–BHR	20
Moroni 2010 [76]	Serum	24	<DL (0.83)	MoM–BHR	15
Moroni 2010 [76]	Serum	56	0.9	MoM–THA	35
Moroni 2010 [76]	Serum	26	0.97	MoM–THA	25
Savarino 2014 [77]	Serum	24	0.89	MoM–BHR	14
Savarino 2014 [77]	Serum	60	0.84	MoM–BHR	19
Savarino 2014 [77]	Serum	108	0.85	MoM–BHR	22

**Table 10 bioengineering-13-00044-t010:** Summary of model performance metrics comparison. Mean absolute error (MAE), mean squared error (MSE), root mean squared error (RMSE), and coefficient of determination (R^2^) are presented.

	MAE	MSE	RMSE	R^2^
Co	0.417	0.329	0.573	0.861
Cr	0.374	0.245	0.495	0.522
Ti	0.233	0.144	0.380	0.707
Mo	0.857	1.483	1.218	0.718
Ni	0.233	0.106	0.325	0.297

**Table 11 bioengineering-13-00044-t011:** Reported cobalt concentration thresholds and reference limits in the literature. Values are drawn from peer-reviewed studies and regulatory guidance, primarily in the context of monitoring patients with hip devices.

Biological Fluid	Co Concentration (μg/L)	Source	Threshold/Limit Type	Context
Blood/Serum	7	MHRA	Threshold	Regulatory level of concern, prompting closer follow-up.
Blood	4.97	Hart 2011 [88]	Threshold	Study-derived cutoff for predicting MoM failure.
Blood	4.5	Sidaginamale 2013 [89]	Threshold	Proposed to indicate abnormal wear in MoM hips.
Serum	4.0 (unilateral)/5.0 (bilateral)	Van Der Straeten 2013 [90]	Acceptable upper limits	Derived in well- vs. poorly functioning hip cohorts.
Not specified (most likely Serum or WB)	3 (upper limit, low risk); 10 (lower limit, high risk)	Kwon 2014 (J Arthroplasty Consensus) [91]	Risk Bands	US consensus for risk stratification; fluid not specified.
Blood/Serum	2–7	EFORT (European [92] Guidelines)	Concern Range	Values for clinical concern, need for additional imaging even in asymptomatic patients.

**Table 12 bioengineering-13-00044-t012:** Reference ranges for Co, Cr, Ti, Ni, and Mo in whole blood and serum as reported by two clinical laboratories.

Element	Fluid	Reference Ranges (µg/L)	Laboratory
Co	Whole Blood	0–0.5	London Health Sciences Centre Laboratories
Cr	Whole Blood	0–1.2	London Health Sciences Centre Laboratories
Ti	Whole Blood	0–3	London Health Sciences Centre Laboratories
Mo	Whole Blood	0–1.6	London Health Sciences Centre Laboratories
Ni	Whole Blood	0–1.3	London Health Sciences Centre Laboratories
Co	Serum	0–0.5	London Health Sciences Centre Laboratories
Cr	Serum	0–0.65	London Health Sciences Centre Laboratories
Ti	Serum	0–3	London Health Sciences Centre Laboratories
Co	Serum	<1.0	Mayo Clinic Laboratory
Ti	Serum	<2.0	Mayo Clinic Laboratory
Mo	Serum	0.3–2.0	Mayo Clinic Laboratory
Ni	Serum	<2.0	Mayo Clinic Laboratory

**Table 13 bioengineering-13-00044-t013:** Multiple studies reporting Co and Cr concentrations with at least one outlier, error bar, or range exceeding the 7 μg/L threshold proposed by the MHRA. Follow-up times indicate the specific timepoint at which reported variability crossed the threshold. Superscript letters correspond to the cited source study.

Metal	Fluid	Follow-Up (Month) with Ranges/Error/Outliers Crossing Threshold	Source Studies
Co	Whole Blood	48 ^a^, 84 ^a^	Bernstein 2012 [96] ^a^
		6 ^b^, 12 ^b^	Bernstein 2012 [96] ^b^
		12 ^c^	Høl 2021 [97] ^c^
		36 ^d^	Forsthoefel 2017 [98] ^d^
		12 ^e^, 24 ^e^	Smolders 2011 [99] ^e^
Cr	Whole Blood	48 ^a^	Bernstein 2012 [96] ^a^
		12 ^b^	Bernstein 2012 [96] ^b^
		12 ^c^, 60 ^c^	Hol 2021 [97] ^c^
		36 ^d^	Forsthoefel 2017 [98] ^d^
		24 ^e^	Smolders 2011 [99] ^e^
Co	Serum	12 ^a^	Garbuz 2010 [100] ^a^
		72–108 ^b^	Koper 2021 [100] ^b^
		6 ^c^	Dahlstrand 2009 [38] ^c^
		3 ^d^, 12 ^d^, 24 ^d^	Witzleb 2006 [39] ^d^
		56 ^e^	Moroni 2010 [76] ^e^
Cr	Serum	132 ^a^, 168 ^a^	Maezawa 2018 [101] ^a^
		24–48 ^b^, 84–108 ^b^	Koper 2021 [102] ^b^
		6 ^c^	Dahlstrand 2009 [38] ^c^
		3 ^d^, 12 ^d^, 24 ^d^	Witzleb 2006 [39] ^d^
		56, ^e^ 58 ^e^	Moroni 2010 [76] ^e^

## Data Availability

The data are contained within the article and Appendix A. The data were collected from previously peer-reviewed and published articles available to the public, as cited in the references.

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
