# Peer review of "Kinetics and Fluid-Specific Behavior of Metal Ions After Hip Replacement"

_bioengineering, 2025, doi:10.3390/bioengineering13010044_

Round 1
Reviewer 1 Report
Comments and Suggestions for Authors
This manuscript presents a comprehensive pooled analysis and kinetic modeling study of metal ion release (Co, Cr, Ti, Mo, Ni) in serum, whole blood, and urine following total hip arthroplasty (THA).
The abstract is structured; instead of "stronger model for cobalt," specify the performance metrics ; check the keywords in accordance with MeSH.
In the introduction please provide more information on the amount and type of hip replacements performed in recent years, by referring to the scientific literature (for e.g. doi: 10.3390/healthcare13192452 ). The aim is well stated at the end.
The methodology for study selection, data extraction, and standardization is clearly described. Please, include a PRISMA-style flowchart to visually depict the process of study identification, screening, eligibility, and inclusion. Also, acknowledge specific potential sources of heterogeneity (e.g., different implant designs [MoM vs. MoP], bearing sizes, patient activity levels) and briefly discuss how the pooling and weighting strategy attempts to mitigate this, while acknowledging it as a key limitation.
In the results the "Model Fit" subsections for each ion (e.g., Serum Co, Whole Blood Co) are somewhat repetitive. Consider creating a single, standardized paragraph structure or a summary table for model diagnostics to improve conciseness. For urinary results (Figs. 5, 9), explicitly state that the data are presented in phases due to a lack of continuous longitudinal reporting, which is a limitation of the available literature. All tables should respect the APA academic format.
The discussion interpret the study results and relate them to other findings from the scientific literature. The ML approach is promising, but please strongly emphasize its exploratory nature.
The conclusions are concise.
Author Response
Reviewer 1
Round 1
Dear Editor,
We sincerely thank the reviewer for their time, feedback, and careful evaluation of our manuscript. Below, we provide responses addressing the comments we received.
Comments
- The abstract is structured; instead of "stronger model for cobalt," specify the performance metrics; check the keywords in accordance with MeSH.
We thank you for your comment. In the abstract, we have added performance metrics (R2 and RMSE) for the RF models for both cobalt and chromium. Additionally, we reviewed and updated keywords and believe they align with MeSH terminology.
- In the introduction, please provide more information on the amount and type of hip replacements performed in recent years, by referring to the scientific literature (for e.g. doi: 10.3390/healthcare13192452 ). The aim is well stated at the end.
Thank you for the positive comment about the aim of the article. We have added more information about the general amount of hip replacements performed in recent years and referred to scientific literature (lines 43-45).
- The methodology for study selection, data extraction, and standardization is clearly described. Please, include a PRISMA-style flowchart to visually depict the process of study identification, screening, eligibility, and inclusion. Also, acknowledge specific potential sources of heterogeneity (e.g., different implant designs [MoM vs. MoP], bearing sizes, patient activity levels) and briefly discuss how the pooling and weighting strategy attempts to mitigate this, while acknowledging it as a key limitation.
We appreciate your constructive suggestion regarding our methodology. We have added a summarized PRISMA-style flow diagram that shows our process of identification, screening/eligibility, and inclusion (now figure 1 at lines 415-420). Additionally, we have added a paragraph acknowledging sources of inter-study heterogeneity, such as bearing size, patient-specific factors, and implant designs. In the same paragraph, we explained how our data pooling and weighting attempts to mitigate this (lines 429-436). This new addition also states how this is an inherent limitation to our methods.
- In the results the "Model Fit" subsections for each ion (e.g., Serum Co, Whole Blood Co) are somewhat repetitive. Consider creating a single, standardized paragraph structure or a summary table for model diagnostics to improve conciseness. For urinary results (Figs. 5, 9), explicitly state that the data are presented in phases due to a lack of continuous longitudinal reporting, which is a limitation of the available literature. All tables should respect the APA academic format.
Thank you for your attention to the repetitiveness of the “Model Fit” portions of the results. We have condensed all model diagnostic explanations into a new subsection “3.4 Nonlinear Model Diagnostics” The write-ups for serum and whole blood Co and Cr have been condensed into a single paragraph each, and we have included the corresponding model outputs into a single, summarized table (now Table 8). We kept serum Ti in its own separate table (Table 9) because an exponential rise-then-decay displays different Best-Fit values compared to the one-phase association seen for Co and Cr.
We also want to thank you for the insight into being more explicit with the use of time phases for urinary data. We have included the statement “Due to lack of longitudinal reporting from data in the literature, time phases were utilized and are defined as Preop” into both urinary bar graphs (now figures 3 and 5).
- The discussion interpret the study results and relate them to other findings from the scientific literature. The ML approach is promising, but please strongly emphasize its exploratory nature.
We thank the reviewer for this valuable comment about the exploratory nature of ML. The discussion on ML has been revised to clarify the exploratory nature of RF. We have added “Since open access cohort data was used, the RF models were not intended to be used as a clinical prediction. Rather the goal was to explore its methodological potential” at lines 1345-1348. We also adjusted other minor portions in discussion section “4.5 Machine Learning Models” in an attempt to not over-claim its use.
Thank you again for your time, consideration, and feedback! We sincerely hope that you find our responses clarifying and satisfactory.
Sincerely,
The Authors

Reviewer 2 Report
Comments and Suggestions for Authors Interesting article on a little covered topic. There are few related articles. My comments: Abstract: Clearly and legibly written. Introduction: Clearly and legibly written. A broad review of the literature related to the research topic. Please include information on all possible metals used in THA, with citations from the literature. Please indicate which metals are most toxic to the body and likely to cause the most complications in patients, with citations from the literature. High-quality, legible tables and figures are required. Clear and well-chosen work goals. Please ad research hypothesis. Materials and methods: Clearly and extensively presented. Please describe precisely and broadly the criteria for inclusion in the study group. Please describe the exclusion criteria from the study in more detail. Please add demographics (race, weight, height, BMI...…). Please add information whether the examined patients had other metal implants in them - for example after fractures - this may distort the results. Good description of the research methodology. Clear and good quality figures. Results: Well written. Well and precisely presented research results. Clear, legible and good quality tables and figures. The discussions: Well written, extensive literature review. Well-described work restrictions. Practical conclusions for anesthesiologists and orthopedists should be added, which result from this work. What possibilities do the results of this work offer in practice? Conclusions: Well presented. They are clear from the results and discussions.Author Response
Reviewer 2
Round 1
Dear Editor,
We sincerely thank the reviewer for their time, feedback, and careful evaluation of our manuscript. Below, we provide responses addressing the comments we received.
Comments
- Abstract: Clearly and legibly written.
We thank the reviewer for the positive comment regarding our abstract and are pleased you found it clear and legible.
- Introduction: Clearly and legibly written. A broad review of the literature related to the research topic. Please include information on all possible metals used in THA, with citations from the literature. Please indicate which metals are most toxic to the body and likely to cause the most complications in patients, with citations from the literature. High-quality, legible tables and figures are required. Clear and well-chosen work goals. Please add research hypothesis.
Thank you for the comments regarding our introduction. We have added information on the main metals used in hip prosthetics in lines 55-67 with citation from the literature. Table 2 also offers a complete metal composition by percentage of the most common surgical grade alloys.
Additionally, we have made it more explicit as to how Co and Cr are relevant to genotoxicity and cytotoxicity and how metal ions are linked to adverse reactions, with references to literature (paragraph starting at line 68). Lastly, we have also added our research hypothesis at line 113 “We hypothesize that Co, Cr, and Ti will show a sharp early postoperative rise with sustained elevation postoperatively and that machine-learning models offers an exploratory approach to characterize temporal trends in the pooled datasets.”
We have also chosen to remove the figure in section 1.1 (previously Figure 1) as we believed this information could be summarized within the text in explaining corrosion-only mechanisms of alloy degradation. Since in vivo conditions include both corrosion and mechanical wear mechanisms, we believe the figure may have been shifting away from the main scope of the paper.
- Materials and methods: Clearly and extensively presented. Please describe precisely and broadly the criteria for inclusion in the study group. Please describe the exclusion criteria from the study in more detail. Please add demographics (race, weight, height, BMI...…). Please add information whether the examined patients had other metal implants in them - for example after fractures - this may distort the results. Good description of the research methodology. Clear and good quality figures.
Thank you for the constructive comments regarding our methodology. In Section 2.3, we have included a PRISMA-style diagram outlining our inclusion and exclusion criteria more explicitly (Figure 1). Also, we have added a brief mention of the demographics for age and BMI, as these were most consistent across retrieved studies (lines 395-396). Not all publication reported a complete set of demographic information, so we displayed a general range across what was available. The distribution of race and ethnicity was inconsistently reported and could not be analyzed quantitatively (paragraph starting at line391).
In the same paragraph above, we have also clarified that most of the available data came from patients with degenerative joint disease states rather than trauma related incidents (rarely reported). When articles specified, bilateral or multi-implant cohorts were excluded to minimize confounding’s.
- Results: Well written. Well and precisely presented research results. Clear, legible and good quality tables and figures.
We sincerely thank the reviewer for this positive feedback about our results and are pleased to hear that our it was clear and organized.
- The discussions: Well written, extensive literature review. Well-described work restrictions. Practical conclusions for anesthesiologists and orthopedists should be added, which result from this work. What possibilities do the results of this work offer in practice?
We want to thank the reviewer for the very insightful comment. Starting at line 1096, we have expanded the discussion to include practical implications for anesthesiologists and surgeons and how the observed trend may help them identify early or late signs of excessive wear or corrosion. We have also mentioned how this knowledge can help inform specialists for potential metal-drug interactions, given our observed kinetic trend. We do recognize that more research should be done to confirm these trends, but if validated, these are the insights that can be drawn.
- Conclusions: Well presented. They are clear from the results and discussions.
We thank the reviewer for the positive comment about our conclusion!
Thank you again for your time, consideration, and feedback! We sincerely hope that you find our responses clarifying and satisfactory.
Sincerely,
The Authors

Reviewer 3 Report
Comments and Suggestions for Authors
Specific Comments
Title
- Clear and representative. No change needed.
Abstract
- Well structured.
- Consider including quantitative results (plateau concentrations, half-time values, R² for Random Forest models) to give readers a concise sense of magnitude.
- Indicate the main conclusion more assertively, e.g., “Serum and whole blood cobalt and chromium displayed distinct kinetic profiles, supporting fluid-specific monitoring strategies.”
Introduction
- Strong review of background literature.
- Could be shortened slightly by merging overlapping descriptions of corrosion mechanisms.
- Add one or two recent references (2023–2024) on implant corrosion and systemic ion toxicity to update the context.
Methods
- Very well documented. Minor clarifications recommended:
- Specify whether the literature search followed PRISMA-style identification and selection flow (include a figure if possible).
- Clarify how duplicate cohorts were handled and whether weighting by cohort size may bias results toward larger series.
- Mention the software version used for GraphPad Prism and Python libraries (scikit-learn, pandas).
- Include details on hyperparameter tuning in the Random Forest model (e.g., grid search, cross-validation).
- Explicitly state the source of each figure/table if adapted from prior publications (ensure permissions if reused).
Results
- Comprehensive and clearly presented.
- Consider summarizing kinetic parameters (plateau, half-time, R²) for all ions in a single summary table for easy comparison.
- Figures are generally high quality, but some captions could more explicitly define abbreviations (e.g., SD, R², Sy.x).
- Report number of included studies and participants by ion type to reinforce transparency.
- In the Random Forest section, mention the total number of data points used for training/testing and the cross-validation split.
Discussion
- Balanced and scientifically sound.
- Suggested improvements:
- Deepen discussion on biological interpretation: why cobalt shows stronger model performance (R² = 0.86) than chromium (R² = 0.52).
- Add a short paragraph on clinical implications: how these kinetics could inform postoperative surveillance and risk stratification.
- Acknowledge that pooled study-level data cannot fully represent individual variability (renal function, implant type, time since surgery).
- Briefly discuss future directions—need for multicenter prospective cohorts using standardized sampling.
Limitations
- Present but could be expanded to include:
- Lack of patient-level data and heterogeneity in assay techniques.
- Potential selection bias in literature inclusion.
- Absence of adjustment for confounders such as implant material composition, patient sex, or renal clearance.
Conclusion
- Concise and supported by findings.
- Consider slightly strengthening clinical take-home message, e.g.:
“Our findings highlight the value of fluid-specific reference kinetics for postoperative monitoring, suggesting that serum cobalt and chromium provide complementary indicators of implant wear and systemic exposure.”
Recommendation of Additional Literature
Impact on Blood Tests of Lower Limb Joint Replacement for the Treatment of Osteoarthritis: Hip and Knee.
Comments on the Quality of English Language- Overall, language is very good.
- Minor suggestions: standardize tense (past for results, present for conclusions), check consistency of µg/L notation, and shorten a few long sentences in Introduction and Discussion.
- Verify all figure legends are self-contained.
Author Response
Reviewer 3
Round 1
Dear Editor,
We sincerely thank the reviewer for their time, feedback, and careful evaluation of our manuscript. Below, we provide responses addressing the comments we received.
- Title: Clear and representative. No change needed.
We appreciate the positive comment regarding our title!
- Abstract: Well structured. Consider including quantitative results (plateau concentrations, half-time values, R² for Random Forest models) to give readers a concise sense of magnitude. Indicate the main conclusion more assertively, e.g., “Serum and whole blood cobalt and chromium displayed distinct kinetic profiles, supporting fluid-specific monitoring strategies.”
We thank the reviewer for the suggestion. We have added quantitative results and RF metrics within the “results” portion of the abstract. Additionally, we have stated our conclusion more assertively by emphasizing the distinct kinetic profiles across ions and their potential use for fluid-specific monitoring.
- Introduction: Strong review of background literature. Could be shortened slightly by merging overlapping descriptions of corrosion mechanisms. Add one or two recent references (2023–2024) on implant corrosion and systemic ion toxicity to update the context.
We deeply appreciate the reviewers suggestion The introduction was modified to remove overlapping information of corrosion mechanisms, and we have included more-recent references from 2022,2023, and 2024). Specifically, they are references [2], [4], and [6]. We have also decided to remove the figure previously inserted in section 1.1 which showed corrosion-only behavior from alloys. We believe that this figure was unnecessary, as in vivo conditions undergo both corrosion and mechanical wear.
- Methods: Very well documented. Minor clarifications recommended: Specify whether the literature search followed PRISMA-style identification and selection flow (include a figure if possible). Clarify how duplicate cohorts were handled and whether weighting by cohort size may bias results toward larger series. Mention the software version used for GraphPad Prism and Python libraries (scikit-learn, pandas). Include details on hyperparameter tuning in the Random Forest model (e.g., grid search, cross-validation). Explicitly state the source of each figure/table if adapted from prior publications (ensure permissions if reused).
We thank the author for the constructive feedback regarding our methodology. In section 2.3, we have included a PRISMA-like figure showing our study selection and identification (Figure 1).
We have also clarified how duplicate cohorts were extremely unlikely, as most studies came from different authors/groups/institutions, although we do note that because this is study-level data this could not be completely ruled out. These statements were included in section 2.4 “Data Extraction and Standardization.”
The methods now shows that ion concentrations were weighted by cohort sample size to balance contribution of small and large groups while acknowledging the exploratory nature of this methodology in capturing broad-scale patterns (section 2.4)
We have added the software versions for GraphPad Prism (v10.6.1), and Python libraries (pandas 1.2.4, numpy 1.20.3, and scikit-learn0.24.1)
Details on hyperparameter tuning in RF model was expanded in the paragraph starting at line 532.
We only extracted numerical data from open access figures (without reusing original figure images in our manuscript) and acknowledge each study in Table 1A. We have added a statement clarifying that all our figures were redrawn and reanalyze by us, the authors (lines 425-426).
- Results: Comprehensive and clearly presented. Consider summarizing kinetic parameters (plateau, half-time, R²) for all ions in a single summary table for easy comparison. Figures are generally high quality, but some captions could more explicitly define abbreviations (e.g., SD, R², Sy.x). Report number of included studies and participants by ion type to reinforce transparency. In the Random Forest section, mention the total number of data points used for training/testing and the cross-validation split.
We appreciate the reviewer’s suggestion regarding our results section. We have added a summary table (table 8) for kinetic parameters comparing serum and WB Co and Cr. Ti had a separate table because it had different parameters and would distort Table 8 if it was included.
We have gone through our figure captions and have revised to define the abbreviations standard deviation (SD), standard deviation of residuals (Sy.x), coefficient of determination (R2), and degrees of freedom (df).
Each of our main figures in the results section has a corresponding table underneath it is providing information about the number of cohorts and reported participants at each follow up interval (preop, 12, 24, 36 months etc.). This was implemented to be transparent as some of the later follow up timepoints have smaller group sizes.
For the RF portion, we have added the training/testing and cross-validation in the methods section, as we believe it fits in the “RF Model Development and Evaluation” subsection (lines 541-550).
No. of training and testing datasets for each ion:
Co= train=64, test=16, Total=80
Cr= train=72, test=18, Total=90
Ti= train=52, test=14, Total=66
Mo=train=28, test=8, Total=36
Ni=train=14, test=4, Total=18
- Discussion: Balanced and scientifically sound. Suggested improvements: Deepen discussion on biological interpretation: why cobalt shows stronger model performance (R² = 0.86) than chromium (R² = 0.52). Add a short paragraph on clinical implications: how these kinetics could inform postoperative surveillance and risk stratification. Acknowledge that pooled study-level data cannot fully represent individual variability (renal function, implant type, time since surgery). Briefly discuss future directions—need for multicenter prospective cohorts using standardized sampling.
We thank the reviewer for the suggested improvements for our discussion section. We have added to our discussion section 4.5 about how the difference in Co and Cr performance may be due to its protein binding, RBC uptake, valency state, and clearance and how Co may demonstrate more consistent behaviors across cohorts.
Also, we expanded on the clinical implications for surgeons, anesthesiologists and pharmacists in section 4.2 of the discussion. In this same section we acknowledge the need for multicenter prospective cohort studies to confirm our results.
- Limitations: Present but could be expanded to include: Lack of patient-level data and heterogeneity in assay techniques. Potential selection bias in literature inclusion. Absence of adjustment for confounders such as implant material composition, patient sex, or renal clearance.
We thank the reviewer for the constructive feedback. Our limitations section (section 4.8) has been expanded and address selection bias and the absence of potential confounders such as implant design, sample size, assay technique, different follow-up intervals, and patient demographics. We also note how the machine learning was intended to be exploratory.
- Conclusion: Concise and supported by findings. Consider slightly strengthening clinical take-home message, e.g.“Our findings highlight the value of fluid-specific reference kinetics for postoperative monitoring, suggesting that serum cobalt and chromium provide complementary indicators of implant wear and systemic exposure.”
We thank the reviewer for the positive comments regarding our conclusion. We have added the statement to strengthen the clinical take home message regarding fluid-specific reference kinetics for postop monitoring.
Thank you again for your time, consideration, and feedback! We sincerely hope that you find our responses clarifying and satisfactory.
Sincerely,
The Authors
